# ELUCIDATING THE DESIGN SPACE OF DEEP STOCHASTIC PROCESSES FOR IMAGE ENHANCEMENT

## ABSTRACT

In this work, we investigate deep stochastic processes for image enhancement. We show that existing approaches can be interpreted as instances of Ornstein–Uhlenbeck processes, diffusion bridges, or diffusion processes, each represented by a stochastic differential equation. As a result, we consolidate 11 methods into a unified mathematical framework and present them in a systematically structured table. This perspective separates the definition of processes from the schedulers and samplers that were originally used. Furthermore, we provide a modular library that implements the proposed methods and facilitates the integration of additional approaches with minimal coding effort. In order to perform comprehensive empirical evaluation among considered approaches, we evaluate them on four image enhancement tasks: super-resolution, colorization, low-light enhancement, and deraining with identical backbones and training protocol ensuring fair and meaningful comparison. The experiments highlight that, while most methods achieve similar results, there are exceptions that make some refinement strategies more effective than others, which we further analyze and explain.

## 1 INTRODUCTION

Image enhancement refers to a broad class of tasks, such as super-resolution, colorization, low-light enhancement, and rain removal, where the goal is to recover a high-quality image from a degraded input. These tasks are inherently generative because the corrupted signal cannot fully determine the high-quality output; the model must synthesize plausible visual details that are missing. As a result, traditional computer vision techniques Danielyan et al. (2011); Guo (2016) typically struggle to produce convincing outputs Chen et al. (2018).

Early breakthroughs in this area came with the introduction of deep learning architectures such as U-Net Ronneberger et al. (2015), which were trained using pixelwise losses, such as mean squared error (MSE) Dong et al. (2014; 2016); Chen et al. (2018). However, these losses proved to be suboptimal because image enhancement is an ill-posed problem—there is usually no single correct solution, but rather a distribution of plausible high-quality reconstructions. Pixelwise losses tend to average over these possibilities, leading to overly smooth and blurry results Ledig et al. (2016).

To address this, the researchers introduced generative adversarial networks (GANs) Ledig et al. (2016); Wang et al. (2018); Nazeri et al. (2018); Jiang et al. (2021), which combine reconstruction or consistency loss with adversarial loss. The adversarial component encourages the model to produce more realistic outputs by penalizing images that the discriminator classifies as unnatural. Although GANs significantly improved perceptual quality over standard methods, they introduced new challenges, most notably unstable and difficult-to-balance training dynamics, that limited their reliability and performance.

To further improve the quality of image enhancement models, researchers have turned to diffusion models Saharia et al. (2022a;b); Jinhui Hou & Yuan (2023). These models have gained popularity due to their strong performance in image generation and relatively stable training behavior. Diffusion models work by gradually removing artificial noise from an image in a series of steps, eventually producing a high-quality output. Although they often achieve impressive results, this approach comes with a tradeoff in speed. Because the denoising process is iterative and typically requires large neural networks, diffusion models are significantly slower than traditional GAN-based methods.

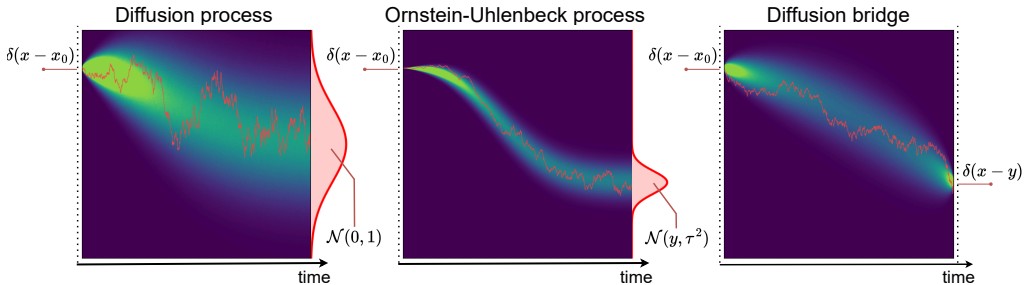

Figure 1: 1D visualization of three classes of considered methods and how conditioning with y affects them. Left: Diffusion models gradually perturb $x_0$ into Gaussian noise $\mathcal{N}(0,1)$, independently of $y$. Middle: Ornstein–Uhlenbeck processes converge to a terminal distribution centered at $y$ with variance $\tau^2$. Right: Diffusion bridges start at $x_0$ and are conditioned to reach $y$ at the terminal time. For ease of visualization, we consider fixed starting point for each process.

More recently, a new class of methods has been proposed specifically for image enhancement Yue et al. (2023b); Delbracio & Milanfar (2023); Li et al. (2023); Liu et al. (2023); Luo et al. (2023a); Zhou et al. (2023b); Yue et al. (2023a); Zhu et al. (2025). Unlike diffusion models, which start from pure Gaussian noise, these methods construct stochastic processes whose forward and reverse trajectories interpolate directly between the distributions of low- and high-quality images. This design better matches the image enhancement setting, as the model evolves from a degraded observation toward its restored version rather than from unrelated random noise.

These refinement processes can be categorized into two generalized families. The first is the Ornstein–Uhlenbeck (OU) process, where the forward process converges to a Gaussian distribution centered on the low-quality image with variance controlled by a temperature parameter. The second is the diffusion bridge, where the process ends exactly at the low-quality image. In Figure 1, we present a visual comparison of these methods together with the standard diffusion process.

Although these approaches are motivated by the structure of image enhancement problems, there has been little empirical work to assess their effectiveness under consistent conditions. In particular, it remains unclear whether they provide measurable advantages over standard diffusion or flow-matching methods that are simply conditioned on the degraded image. The goal of this work is to establish a unified framework for these methods and to systematically compare their performance using identical architectures and training protocols, ensuring fair and meaningful evaluation.

The main contributions of our work can be summarized as follows: (i) We unified the definition of the most widely known deep stochastic processes from the literature. The design choices across these methods are analyzed and presented in structured, easy-to-read tables. (ii) We conducted a comprehensive set of experiments across several image enhancement tasks to evaluate and compare the performance of prominent approaches in consistent and fair settings. (iii) We developed a modular library that unifies recent state-of-the-art deep stochastic process methods for image enhancement by integrating their formulations together with schedulers, samplers, and time discretization techniques, enabling flexible model composition and easy extension with minimal code.

## 2 UNIFIED FRAMEWORK

In this section, we present popular image enhancement methods by framing them as continuous stochastic processes. We start by introducing the general mathematical framework. After that, we examine 11 methods from the literature, grouped into three categories: diffusion models, Ornstein–Uhlenbeck processes, and diffusion bridges. Lastly, we introduce library that implements all unified methods. We highlight our major contributions with propositions and remarks.

### 2.1 GENERAL DEFINITIONS

We define each method with three components: forward process, transition kernels, and base distribution.

Table 1: Different design choices of forward SDE and transition densities of different iteration-based methods. We organized the methods into three groups: diffusion processes, OU processes and diffusion bridges. We denote $\tau$ as temperature, $\gamma$ as SOC penalty, $\beta_t$ as a noise scheduler, its Riemann integral $\alpha_{s,t} = \int_s^t \beta_z dz$, and $\phi_{s,t} = \exp(-\alpha_{s,t})$. We shorten the notation by $\alpha_t \equiv \alpha_{0,t}$, $\phi_t \equiv \phi_{0,t}$. For original choices of $\beta_t$, $\tau$, or $\gamma$ for each method please see Table 3.

| Methods | Forward SDE $\mathrm{d}\mathbf{x}_t = \mathbf{f}(\mathbf{x}_t, t, \mathbf{y})\,\mathrm{d}t + g(t)\,\mathrm{d}\mathbf{w}_t$ | | Transition kernel $p_t(\mathbf{x}_t|\mathbf{x}_0, \mathbf{y}) = \mathcal{N}(\mathbf{x}_t; \boldsymbol{\mu}_t(\mathbf{x}_0, \mathbf{y}), \sigma_t^2 \boldsymbol{I})$ | | Base dist. |
|---|---|---|---|---|---|
| | $\mathbf{f}(\mathbf{x}_t, t, \mathbf{y})$ | $g^2(t)$ | $\boldsymbol{\mu}_t(\mathbf{x}_0, \mathbf{y})$ | $\sigma_t^2$ | $p_1(\mathbf{x}_1|\mathbf{y})$ |
| DM-VE | $0$ | $\beta_t$ | $\mathbf{x}_0$ | $\alpha_t$ | $\mathcal{N}(\mathbf{x}_0, \alpha_1 \boldsymbol{I})$ |
| DM-VP | $-\beta_t \mathbf{x}_t$ | $2\beta_t$ | $\phi_t \mathbf{x}_0$ | $1 - \phi_t^2$ | $\mathcal{N}(\mathbf{0}, \boldsymbol{I})$ |
| FM | | $2(1-\phi_t)\beta_t$ | | $(1-\phi_t)^2$ | |
| IR-SDE | $\beta_t(\mathbf{y}-\mathbf{x}_t)$ | $2\tau^2\beta_t$ | $\phi_t \mathbf{x}_0 + (1-\phi_t)\mathbf{y}$ | $\tau^2(1-\phi_t^2)$ | $\mathcal{N}(\mathbf{y}, \tau^2\boldsymbol{I})$ |
| ResShift | | $\tau^2(2-\phi_t)\beta_t$ | | $\tau^2(1-\phi_t)$ | |
| InDI | | $2\tau^2(1-\phi_t)\beta_t$ | | $\tau^2(1-\phi_t)^2$ | |
| BBDM / DDBM-VE / I$^2$SB | $\frac{\beta_t}{\alpha_{t,1}}(\mathbf{y}-\mathbf{x}_t)$ | $\beta_t$ | $\frac{\alpha_{t,1}}{\alpha_1}\mathbf{x}_0 + \frac{\alpha_t}{\alpha_1}\mathbf{y}$ | $\frac{\alpha_t \alpha_{t,1}}{\alpha_1}$ | $\delta(\mathbf{x}-\mathbf{y})$ |
| DDBM-VP | $\beta_t\left(\frac{2\phi_{t,1}}{1-\phi_{t,1}^2}\mathbf{y} - \frac{1+\phi_{t,1}^2}{1-\phi_{t,1}^2}\mathbf{x}_t\right)$ | $2\beta_t$ | $\phi_t \frac{1-\phi_{t,1}^2}{1-\phi_1^2}\mathbf{x}_0 + \phi_{t,1}\frac{1-\phi_t^2}{1-\phi_1^2}\mathbf{y}$ | $\frac{(1-\phi_{t,1}^2)(1-\phi_t^2)}{1-\phi_1^2}$ | |
| GOUB | $\beta_t \frac{1+\phi_{t,1}^2}{1-\phi_{t,1}^2}(\mathbf{y}-\mathbf{x}_t)$ | $2\tau^2\beta_t$ | $\phi_t \frac{1-\phi_{t,1}^2}{1-\phi_1^2}\mathbf{x}_0 + (1-\phi_t\frac{1-\phi_{t,1}^2}{1-\phi_1^2})\mathbf{y}$ | $\tau^2\frac{(1-\phi_{t,1}^2)(1-\phi_t^2)}{1-\phi_1^2}$ | |
| UniDB$^\dagger$ | $\beta_t \frac{(\gamma\tau^2)^{-1}+1-\phi_{t,1}^2}{(\gamma\tau^2)^{-1}+1+\phi_{t,1}^2}(\mathbf{y}-\mathbf{x}_t)$ | | $\phi_t \frac{\gamma^{-1}+1-\phi_{t,1}^2}{\gamma^{-1}+1-\phi_1^2}\mathbf{x}_0 + (1-\phi_t\frac{\gamma^{-1}+1-\phi_{t,1}^2}{\gamma^{-1}+1-\phi_1^2})\mathbf{y}$ | | |

$^\dagger$ We consider UniDB that modifies GOUB, which was also used as a main example in the original work.

**Forward process** denoted as $(\mathbf{x}_t \in \mathbb{R}^d)_{t\in[0,1]}$ that follows a stochastic differential equation (SDE) of the general form:

$$\mathrm{d}\mathbf{x}_t = \mathbf{f}(\mathbf{x}_t, t, \mathbf{y})\,\mathrm{d}t + g(t)\,\mathrm{d}\mathbf{w}_t, \qquad (1)$$
$$\mathbf{x}_0 \sim p_{data}(\mathbf{x}_0), \quad \mathbf{y} \sim p_{low}(\mathbf{y}|\mathbf{x}_0),$$

where $\mathbf{w}_t \in \mathbb{R}^d$ is a d-dimensional Wiener process, $\mathbf{y} \in \mathbb{R}^d$ is $d$-dimensional random variable denoting corresponding low-quality images, $\mathbf{f} : \mathbb{R}^d \times [0,1] \times \mathbb{R}^d \to \mathbb{R}^d$ is a deterministic drift, and $g : [0,1] \to \mathbb{R}$ is a diffusion coefficient, and these two functions completely define a model. The model needs to be constructed so that at the terminal time $t = 1$ the process converges to its base distribution $p_1$ which should be easy to sample from. Note that $\mathbf{y}$ depends only on $\mathbf{x}_0$ and $\mathbf{x}_0$ is independent of Brownian motion, therefore, we can still consider equation 1 in the Ito sense.

**Transition kernel** that tells what is the solution to the corresponding forward SDE given $\mathbf{x}_0$ and $\mathbf{y}$. The transition kernel formula is defined as:

$$p(\mathbf{x}_t|\mathbf{x}_0, \mathbf{y}) = \mathcal{N}(\mathbf{x}_t; \boldsymbol{\mu}_t(\mathbf{x}_0, \mathbf{y}), \sigma_t^2 \boldsymbol{I}), \qquad (2)$$

with specific definitions of the functions $\boldsymbol{\mu}_t$ and $\sigma_t^2$ for each method.

**Base distribution** that can be treated as a transition kernel at the terminal time $t = 1$. The base distribution is indicated as $p_1(\mathbf{x}_1|\mathbf{y})$ and it can be the Gaussian or Dirac delta depending on the method.

In order to generate a new high-quality image $\hat{\mathbf{x}}$, we need to first sample $\mathbf{x}_1 \sim p_1(\mathbf{x}_1|\mathbf{y})$ and reverse equation 1. Thanks to Anderson (1982); Zhang & Chen (2021) we have another SDE, where time

goes backward and it almost surely goes through $\mathbf{x}_0$ at time $t = 0$

$$d\mathbf{x}_t = [\mathbf{f}(\mathbf{x}_t, t, \mathbf{y}) - \frac{\lambda^2 + 1}{2} g(t)^2 \nabla_{\mathbf{x}_t} \log p_t(\mathbf{x}_t | \mathbf{y})] dt + \lambda g(t) d\bar{\mathbf{w}}_t, \tag{3}$$

$$\mathbf{x}_1 \sim p_1(\mathbf{x}_1 | \mathbf{y}), \quad \mathbf{y} \sim p_{low}(\mathbf{y}),$$

where $\bar{\mathbf{w}}_t \in \mathbb{R}^d$ is a d-dimensional reversed Wiener process and $\lambda \in \mathbb{R}^+ \cup \{0\}$ is a nonnegative parameter that controls the level of stochasticity of the reverse process. In principle, equation 3 can be solved numerically using standard samplers such as Euler–Maruyama. In practice, however, this requires access to $\log p_t(\mathbf{x}_t | \mathbf{y})$, which is intractable since it involves integration over $p_{data}$. Instead, we approximate its gradient with a neural network $v_\theta$ with learnable parameters $\theta$ such that:

$$\theta^* = \arg\min_\theta \mathbb{E} ||v_\theta(\mathbf{x}_t, t, \mathbf{y}) - \nabla_{\mathbf{x}_t} \log p_t(\mathbf{x}_t | \mathbf{x}_0, \mathbf{y})||^2, \tag{4}$$

which is known as the score matching loss function Hyvärinen & Dayan (2005); Vincent (2011); Song & Ermon (2019).

One of the main contributions of our work is Table 1, where we provide the unified definitions for each of the methods. The columns are divided into three sections that describe the SDE of the forward process, transition kernels, and base distributions. The methods are divided into three groups: diffusion models, OU processes, and diffusion bridges.

Each model is defined independently of the chosen scheduler $(\beta_t > 0)_{t \in [0,1]}$. We denote the Riemann integral of the scheduler as $\alpha_{s,t} = \int_s^t \beta_s ds$ and $\phi_{s,t} = \exp(-\alpha_{s,t})$. For convenience, we shorten $\alpha_t \equiv \alpha_{0,t}$ and $\phi_t \equiv \phi_{0,t}$. The temperature is denoted as $\tau$ in all the methods. For the parameters of each method, such as the scheduler, sampler, or temperature value, see Table 3.

## 2.2 DIFFUSION MODELS

This class of models were initially designed to be unconditional, therefore $\mathbf{y}$ is not used in the definitions of the forward processes, transition kernels or base distributions. To make the model conditional and generate the enhanced version of the input $\mathbf{y}$, one must add $\mathbf{y}$ as an additional input to the backbone $v_\theta$ Rombach et al. (2022); Saharia et al. (2023); Jinhui Hou & Yuan (2023).

**Diffusion model** (DM) Sohl-Dickstein et al. (2015); Ho et al. (2020) was originally introduced as a discrete Markov process that gradually transforms the data distribution into Gaussian noise. Later, Song et al. (2020b) reformulated it in terms of two continuous processes: Variance Exploding (VE) and Variance Preserving (VP), summarized in Table 1. In VE, the expected value of the transition kernel stays constant, while the variance grows to infinity. In VP, the definition of drift $\mathbf{f}(\mathbf{x}_t, t) = -\beta_t \mathbf{x}_t$ pushes the $\mathbf{x}_t$ towards zero with a force proportional to the scheduler $\beta_t$. This force counteracts the Wiener process and ensures that the stationary distribution is standard Gaussian $\mathcal{N}(\mathbf{0}, \boldsymbol{I})$. Originally, the authors used the linear scheduler with parameters $\beta_{max} = 10$ and $\beta_{min} = 0.05$[1].

**Flow Matching** Lipman et al. (2022); Liu et al. (2022), derived from Normalizing Flow theory, can be viewed as deterministic vector fields that transport one distribution to another. Their design has been shown to achieve strong performance, and therefore they are important to include in our comparison.

**Proposition 2.1.** *Let $\mathbf{x}_t$ be a continuous stochastic process that follows given SDE*

$$d\mathbf{x}_t = -\beta_t \mathbf{x}_t dt + \sqrt{2(1 - \phi_t)\beta_t} d\mathbf{w}_t. \tag{5}$$

*If the scheduler is defined as $\beta_t = \frac{1}{1-t}$, then for any $\mathbf{x}_0 \sim p_{data}(\mathbf{x}_0)$ and $\epsilon \sim \mathcal{N}(\mathbf{0}, \boldsymbol{I})$ we have $\mathbf{x}_t = (1 - t)\mathbf{x}_0 + t\epsilon$ for $t \in [0, 1)$, which corresponds to transitions of Liu et al. (2022).*

The proof is in Appendix A.1. With proposition 2.1 we can reverse equation 5 together with Euler sampling to recover the original implementation of Liu et al. (2022)[2] we can also sample the reverse of this process using any other standard sampler. We include this method in the comparison to measure its effectiveness against conditioned processes.

---

[1] You might encounter $\beta_{max} = 20$, $\beta_{min} = 0.1$, but then the SDE is given as $d\mathbf{x}_t = -\frac{1}{2}\beta_t \mathbf{x}_t dt + \sqrt{\beta_t} d\mathbf{w}_t$ which is equivalent. We chose these values to highlight the connection with Ornstein-Uhlenbeck process.

[2] We do not consider rectification procedure.

## 2.3 ORNSTEIN-UHLENBECK PROCESSES

The second group considered in our framework is a generalized Ornstein-Uhlenbeck process with formula

$$d\mathbf{x}_t = \beta_t(\boldsymbol{\mu} - \mathbf{x}_t)\,dt + g(t)\,d\mathbf{w}_t, \tag{6}$$

where $\beta_t > 0$ and $g(t) \geq 0$ are functions of time $t$. The process converges to a Gaussian stationary distribution with mean $\boldsymbol{\mu}$ and variance determined by $g(t)$. It is easy to see that the diffusion and flow matching models are also Ornstein-Uhlenbeck processes with $\boldsymbol{\mu} = \mathbf{0}$. In this section, we consider OU processes with stationary distribution $p_1(\mathbf{x}_1|\mathbf{y}) = \mathcal{N}(\mathbf{x}_1; \mathbf{y}, \tau^2\boldsymbol{I})$, therefore, we denote $\mathbf{y} \equiv \boldsymbol{\mu}$ hereafter.

**IR-SDE** Luo et al. (2023a) is the most standard version of the OU process with a diffusion coefficient $g(t)$ similar to the diffusion model. In the original implementation, the authors used a numerically calculated cosine scheduler for a given number of steps. We generalized it to a continuous function that works with any number of steps without the need for recalculation (see Appendix B).

**ResShift** Yue et al. (2023b) was originally proposed as a discrete model based on the Markov chain that interpolates between high-quality image $\mathbf{x}_0$ and Gaussian centered in low-quality $\mathbf{y}$ with specified variance $\tau^2$. The one-step forward equation is written as

$$p(\mathbf{x}_t|\mathbf{x}_{t-1}, \mathbf{y}) = \mathcal{N}(\mathbf{x}_t; \mathbf{x}_{t-1} + \delta_t\mathbf{e}, \tau^2\delta_t\boldsymbol{I}), \tag{7}$$

where $\mathbf{e}$ means residual between low- and high-quality images, i.e. $\mathbf{e} = \mathbf{y} - \mathbf{x}_0$, $\delta_t$ is an arbitrary noise scheduler for a discrete process, and time $t$ is a natural number between 0 and some sufficiently large $T$. Notably, we can also write its marginalized equation that transitions from $\mathbf{x}_0$ to $\mathbf{x}_t$ at arbitrary timestep $t$ within one step

$$p(\mathbf{x}_t|\mathbf{x}_0, \mathbf{y}) = \mathcal{N}(\mathbf{x}_t; \mathbf{x}_0 + \eta_t\mathbf{e}, \tau^2\eta_t\boldsymbol{I}). \tag{8}$$

Here, $\eta_t$ is cumulative sum of previous deltas, $\eta_t = \sum_{i=0}^{t}\delta_i$.

Let us define $\phi_t = 1 - \eta_t$ and rewrite $\mathbf{e}$ as $\mathbf{y} - \mathbf{x}_0$

$$p(\mathbf{x}_t|\mathbf{x}_0, \mathbf{y}) = \mathcal{N}(\mathbf{x}_t; \phi_t\mathbf{x}_0 + (1 - \phi_t)\mathbf{y}, \tau^2(1 - \phi_t)\boldsymbol{I}). \tag{9}$$

With this form we are looking for SDE that produces such transitions.

**Proposition 2.2.** *The OU-process with the following stochastic differential equation*

$$d\mathbf{x}_t = \beta_t(\mathbf{y} - \mathbf{x}_t)\,dt + \tau\sqrt{\beta_t(2 - \phi_t)}\,d\mathbf{w}_t \tag{10}$$

*shares the same transition densities as the original ResShift model from equation 9 for $t \in [0, 1]$.*

The proof is in Appendix A.2. With proposition 2.2 we can use exponential scheduler and perform ancestral sampling on equation 3 with $\lambda = 1$ to recover the original implementation.

**InDI** Delbracio & Milanfar (2023) defined a continuous process with the following formula for intermediate samples

$$\mathbf{x}_t = (1 - t)\mathbf{x}_0 + t\mathbf{y} + \tau t\epsilon, \quad \epsilon \sim \mathcal{N}(\mathbf{0}, \boldsymbol{I}). \tag{11}$$

The generative process starts with $\mathbf{x}_1 \sim \mathcal{N}(\mathbf{x}_1; \mathbf{y}, \tau^2\boldsymbol{I})$ and solves the deterministic ODE using the Euler method.

**Proposition 2.3.** *The OU-process with the following stochastic differential equation*

$$d\mathbf{x}_t = \beta_t(\mathbf{y} - \mathbf{x}_t)\,dt + \tau\sqrt{2\beta_t(1 - \phi_t)}\,d\mathbf{w}_t. \tag{12}$$

*If the scheduler is defined as $\beta_t = \frac{1}{1-t}$, then for any $\mathbf{x}_0 \sim p_{data}(\mathbf{x}_0)$ and $\epsilon \sim \mathcal{N}(\mathbf{0}, \boldsymbol{I})$ we have the same marginals as equation 11 for $t \in [0, 1)$.*

The proof is in Appendix A.3. With proposition 2.3 we can use the reversed scheduler $\beta_t = \frac{1}{1-t}$ with the Euler sampler of reversed equation 3 with $\lambda = 0$ to recover the original implementation. Similarly to previous propositions, the choice of sampler is independent of the model definition, and therefore we evaluate several variants.

## 2.4 DIFFUSION BRIDGES

The final class of models modifies the forward SDE to ensure that the process reaches $\mathbf{y}$ at the terminal time $t = 1$ almost surely. For any SDE with a form

$$\mathrm{d}\mathbf{x}_t = \mathbf{f}(\mathbf{x}_t, t)\,\mathrm{d}t + g(t)\,\mathrm{d}\mathbf{w}_t, \tag{13}$$

one can apply the h-transform Doob & Doob (1984) to get conditioned SDE in a form

$$\mathrm{d}\mathbf{x}_t = [\mathbf{f}(\mathbf{x}_t, t) + g^2(t)\nabla_{\mathbf{x}_t} \log p_1(\mathbf{y}|\mathbf{x}_t)]\,\mathrm{d}t + g(t)\,\mathrm{d}\mathbf{w}_t. \tag{14}$$

**BBDM** Li et al. (2023) uses the definition of the Brownian bridge, summarized in Table 1.

**Remark 2.1.** *The Brownian bridge definition can be seen as the result of applying the h-transform on the Wiener process. The Wiener process can be generalized to the diffusion VE variant with a constant scheduler $\beta_t = 1$.*

**DDBM-VE & DDBM-VP** Zhou et al. (2023b) generalized the idea of applying the h-transform to the VE and VP variants of diffusion models, which gives the possibility of using an arbitrary scheduler. Notably, with Remark 2.1 for DDBM-VE if we set $\beta_t = 1$ we recover BBDM, therefore, these models have the same definition but different scheduler choice. DDBM-VP is the only conditional method considered that its expected value at any time is not the interpolation between a clean image and a low-quality input, because during the trajectory $\mathbb{E}[\mathbf{x}_t]$ tends to $\mathbf{0}$ with force proportional to the scheduler $\beta_t$.

**I²SB** Liu et al. (2023) approached the bridge problem from a different perspective. Their theory is rooted in the findings on dynamic Schrodinger bridges (SB) from Chen et al. (2021), but they considered the scenario, where we can sample from the joint distribution $p_{data,low}(\mathbf{x}, \mathbf{y})$ during training, which is generally not required for SB models.

**Remark 2.2.** *Deriving the bridge process from Liu et al. (2023) Theory 3.1 with $\mathbf{f}(\mathbf{x}_t, t) = \mathbf{0}$ and $g(t) = \sqrt{\beta_t}$ leads to the same result as applying h-transform Doob & Doob (1984) to the exact same SDE $\mathrm{d}\mathbf{x}_t = \sqrt{\beta_t}\,\mathrm{d}\mathbf{w}_t$. This gives the definition of DDBM-VE Zhou et al. (2023b).*

We elaborate on Remark 2.2 in Appendix A.4. In I²SB, the authors designed a symmetric quadratic scheduler to keep the transition variance symmetric in the interval $t \in [0, 1]$ (see Appendix B).

**GOUB** Yue et al. (2023a) applied h-transform on OU process with the same parameterization as IR-SDE. They additionally introduce mean-reverting ODE, that simply omits the Brownian motion part in equation 3 with $\lambda = 1$ which yields state-of-the-art PSNR and SSIM results in deraining task on Rain100H benchmark. However, this process does not have the same marginals as the forward process and, by applying twice as much influence to score, the process naturally collapses to be closer to expected value, resulting in oversmoothing, as the LPIPS and FID metrics are worse than the standard samplers used along with the same model (see Yue et al. (2023a) Tables 1-3, as well as Tables 9 and 10 in Appendix).

**UniDB** Zhu et al. (2025) generalizes the diffusion bridges using stochastic optimal control (SOC) theory. The goal is to find a controller $\mathbf{u}_{t,\gamma}^*$, such that it minimizes the following linear quadratic SOC problem

$$\mathbf{u}_{t,\gamma}^* = \arg\min_{\mathbf{u}_{t,\gamma} \in \mathcal{U}} \mathbb{E}\left[\int_0^1 \frac{1}{2}||\mathbf{u}_{t,\gamma}||_2^2 dt + \frac{\gamma}{2}||\mathbf{x}_1^u - \mathbf{y}||_2^2\right], \tag{15}$$

$$\text{st. } \mathrm{d}\mathbf{x}_t^u = [\mathbf{f}(\mathbf{x}_t^u, t, \mathbf{y}) + g(t)\mathbf{u}_{t,\gamma}]\,\mathrm{d}t + g(t)\,\mathrm{d}\mathbf{w}_t, \quad \mathbf{x}_0^u = \mathbf{x} \sim p_{data}(\mathbf{x}). \tag{16}$$

Here, $\mathbf{x}_t^u$ denotes the stochastic process under the control of $\mathbf{u}_{t,\gamma}$, $||\mathbf{u}_{t,\gamma}||_2^2$ is an instantaneous cost at time $t$, and $\frac{\gamma}{2}||\mathbf{x}_1^u - \mathbf{y}||_2^2$ is a terminal cost with a penalty $\gamma$. If $\gamma \to \infty$, we only optimize the terminal cost that gives us $\mathbf{x}_1^u = \mathbf{y}$ which is equivalent to applying h-transform. However, for a finite positive $\gamma$ we can trade off two costs. In particular, the authors showed that equation 15 can be solved analytically for all diffusion bridges considered and as an example they described and trained the modified GOUB model with different $\gamma$ values with $1\mathrm{e}4^3$ proven to give the best LPIPS and FID results. We follow Zhu et al. (2025) with the same choices of the modified method and $\gamma$ in our experiments.

---

[3]In the original paper it was $\gamma = 1\mathrm{e}7$ but we need to normalize it with our chosen terminal time $T = 1$ instead $T = 1000$.

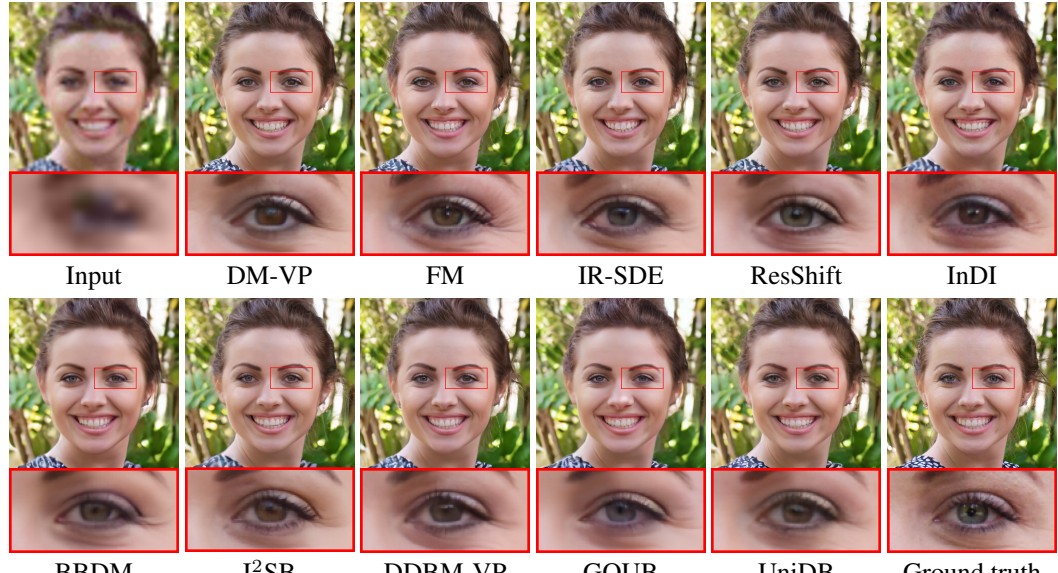

Figure 2: Visual comparison of image super resolution results performed on FFHQ dataset. Results were generated using ancestral sampling with 35 steps. All methods achieve similar visual quality, except BBDM and GOUB, which produce slightly blurred outputs.

## 2.5 LIBRARY

Building on the unified mathematical framework that underlies all considered methods, we release *Ito Vision*, a Python library implemented in PyTorch. The library combines the tested approaches into a single modular framework. This ensures consistency with the theoretical formulation and enables faithful reproduction of our results. Beyond serving as a reference implementation, *Ito Vision* is designed to be easily extensible: by following the unified framework summarized in Table 1, researchers can prototype and integrate new methods with minimal effort. We hope this library will provide a practical foundation for future research and accelerate the development of novel approaches in this domain. Further details are provided in Appendix C.

## 3 EXPERIMENTS

In this section, we report experiments on four image enhancement tasks: super-resolution, low-light enhancement, colorization, and deraining. We first describe the training setup, then compare the methods, evaluate different samplers and discretization techniques, and finally discuss the results.

**Experimental setup** For different tasks, we used backbones of different sizes to test the scalability of the methods. However, within each task, all models share the same backbone size, architecture, and training parameters to ensure a fair comparison. If the method has some additional parameters, such as temperature, we evaluated it with its default settings from the original work, except for network parameterization, where we found that the prediction of $\mathbf{x}_0$ performs best for all methods. We use MSE as a common loss function. For single-image super-resolution, we used the FFHQ dataset Karras et al. (2019) downsampled by a factor of $8$ ($64 \times 64 \rightarrow 512 \times 512$). Low-light image enhancement was conducted on the LOL dataset Wei et al. (2018), image colorization on ImageNet Russakovsky et al. (2015) with latent diffusion setup, and image deraining on Rain1400 Fu et al. (2017). A detailed description of the setup for each experiment is provided in Appendix D and Table 4.

**Experimental results** We compare diffusion, OU processes, and diffusion bridges and present the results in Table 2 (along with 5, 6, 7, and 8 with more details). The key insight from these experiments is that all the methods achieve very similar performance in all tasks. However, we noticed some trends. First, InDI yields the worst LPIPS scores among all OU processes. We explore this

Table 2: Results on four selected tasks using ancestral sampling with Network Forward Evaluations (NFE) = 35. The best values are **bolded** and second to best are underscored. We do not notice any significant improvements of conditional processes over standard diffusion. In fact, in super-resolution and low-light image enhancement diffusion achieves best results.

| Model | Super-Resolution | | | Low-light Enhancement | | | Colorization | | | Deraining | | |
|---|---|---|---|---|---|---|---|---|---|---|---|---|
| | PSNR | SSIM | LPIPS | PSNR | SSIM | LPIPS | PSNR | SSIM | LPIPS | PSNR | SSIM | LPIPS |
| DM-VP | **27.86** | **0.748** | **0.181** | 23.01 | 0.685 | **0.186** | 21.73 | 0.685 | 0.191 | 29.02 | 0.853 | 0.055 |
| FM | 27.15 | 0.730 | 0.183 | 22.57 | 0.678 | 0.209 | 21.64 | 0.685 | 0.191 | 28.89 | 0.844 | 0.057 |
| IR-SDE | 26.97 | 0.718 | 0.184 | 23.07 | 0.679 | 0.237 | 22.02 | 0.691 | 0.187 | 28.15 | 0.827 | 0.061 |
| ResShift | 27.66 | 0.746 | 0.184 | 23.15 | **0.693** | 0.195 | 21.82 | 0.689 | 0.189 | 29.17 | 0.856 | 0.052 |
| INDI | 27.21 | 0.713 | 0.199 | 21.48 | 0.633 | 0.323 | 22.28 | 0.699 | 0.191 | 28.17 | 0.816 | 0.069 |
| BBDM | 27.14 | 0.728 | 0.252 | 22.84 | 0.687 | 0.191 | 22.17 | 0.698 | 0.182 | **30.10** | **0.873** | **0.049** |
| DDBM-VE | 27.11 | 0.724 | 0.186 | **23.18** | 0.680 | 0.224 | 22.36 | 0.700 | **0.181** | 28.65 | 0.837 | 0.054 |
| DDBM-VP | 27.46 | 0.741 | 0.190 | 22.55 | 0.659 | 0.231 | 22.23 | 0.697 | 0.184 | 28.94 | 0.842 | 0.052 |
| $I^2$SB | 27.46 | 0.733 | 0.184 | 21.86 | 0.655 | 0.245 | **22.39** | **0.701** | 0.182 | 29.29 | 0.854 | 0.050 |
| GOUB | 27.24 | 0.735 | 0.224 | 22.57 | 0.669 | 0.233 | 21.59 | 0.681 | 0.193 | 28.38 | 0.831 | 0.061 |
| UniDB | 27.59 | 0.739 | **0.181** | 22.80 | 0.679 | 0.223 | 21.69 | 0.684 | 0.190 | 28.02 | 0.824 | 0.060 |

phenomenon and propose a possible explanation which we discuss in the next paragraph. Moreover, we note that UniDB improves the performance of GOUB with respect to the LPIPS metric, which might suggest that training the model on shorter trajectories might help stay in the well-modeled regions.

The visual comparisons in Figures 2, 6, 7, and 8 further confirm this central result that once trained under the same protocol, the methods produce nearly indistinguishable outputs. In fact, in many cases, the plain diffusion baselines surpass the more elaborate variants, suggesting that the choice of process offers limited practical benefit.

In addition, we study several samplers to determine which work best with our considered methods. The results shown in Tables 9 and 10 reveal that ancestral sampling is the most consistent method for all models. The Euler method on reversed ODE works very well for diffusion and OU processes. For all diffusion bridges, deterministic samplers (e.g., Euler ODE, Exponential Integrator ODE, mean-reverting ODE, and second-order Runge-Kutta methods) yield high LPIPS scores ($> 0.3$), suggesting that these methods perform best when combined with stochastic samplers. This is because trajectories start from fixed points and the only source of stochasticity in these models is the Brownian motion, which deterministic samplers omit. As a result, the trajectories follow straight lines between $\mathbf{y}$ and $\mathbf{x}_0$, converging to the averaged ground truth (oversmoothing problem).

Finally, we conducted a series of experiments with different discretization techniques visualized in Figure 4. We show the results in Table 11, however, we find that this choice does not influence the quality of the final samples.

**Discussion** Methods based on the OU process or diffusion bridge were proposed with the following rationale: when starting from a low-quality image, the network already has some structural information about the scene and therefore does not have to cover as much space as standard diffusion, but it can traverse between two distributions and focus on recovering details. However, we find no evidence that narrowing the modeled space improves the results. Moreover, we suggest that it is the opposite, that this inductive bias can make the model less effective. By setting the temperature to very small values (e.g. $\tau = 0.06$ for InDI), the model learns that the trajectories are almost straight, and thus it learns to extrapolate $\mathbf{x}_t - \mathbf{y}$ vector[4]. We further analyze this phenomenon and measure the co-linearity of $\mathbf{y}$, $\mathbf{x}_t$, and $\mathbf{x}_0$ at inference and compare it with the LPIPS of the sequence of model predictions $\hat{\mathbf{x}}_{0|t}$. As shown in Figure 5, low-temperature models show much higher collinearity and slower LPIPS improvement, especially for small $t$, where most of the details are believed to be generated Karras et al. (2022). We can see the disproportion of the temperature for the OU processes in Figure 3.

Increasing the level of noise during training mitigates this effect, as the model can no longer learn to extrapolate, but rather learns to guess where $\mathbf{x}_0$ can be, improving its ability to recover accurate estimates. However, by increasing temperature, the process loses information about the initial signal and becomes a pure diffusion model, where $\mathbf{y}$ has no impact on the trajectory.

---

[4]This is because the backbone $v_\theta$ takes $\mathbf{x}_t$, $t$ and $\mathbf{y}$ as parameters, so it can deduce the exact data point $\mathbf{x}_0$ provided $\mathbf{x}_t$ has no noise.

## 4 RELATED WORKS

**SP for image enhancement** Diffusion models were first introduced for unconditional generation Sohl-Dickstein et al. (2015); Song & Ermon (2019); Ho et al. (2020); Song et al. (2020b) and later adapted to image enhancement by concatenating the corrupted image with the input Saharia et al. (2022a;b); Rombach et al. (2022); Jinhui Hou & Yuan (2023). Another line of work Wang et al. (2022); Yang et al. (2023) applies guidance to recover a clean version of the low-quality image. Following Bansal et al. (2023), which showed that the forward process does not have to be Gaussian, the researchers proposed alternative formulations that directly incorporate input into the diffusion process. Examples include ResShift Yue et al. (2023b), which defines a Markov chain converging to a Gaussian centered on the low-quality image; InDI Delbracio & Milanfar (2023), which adapts flow matching Lipman et al. (2022); Liu et al. (2022) using the low-quality image distribution as the base; IR-SDE Luo et al. (2023a), which introduces a mean-reverting Ornstein–Uhlenbeck process; I$^2$SB Liu et al. (2023), derived from the Schrödinger Bridge theory Chen et al. (2021); De Bortoli et al. (2021; 2024); DDBM and GOUB Zhou et al. (2023b); Yue et al. (2023a), based on Doob's h-transform Doob & Doob (1984); and UniDB Zhu et al. (2025), which is based on stochastic optimal control (SOC). The goal of this paper is to provide a unified formulation for these methods.

**Methods not included in our studies** We exclude methods that use the definition of the degradation function Kawar et al. (2021; 2022); Chung et al. (2022); Song et al. (2023). We do not consider methods that improve the architectures or samplers of our chosen methods, such as Refusion Luo et al. (2023b) that focuses on the design of the backbone for the IR-SDE method or I$^3$SB Wang et al. (2025) that applies DDIM Song et al. (2020a) sampling to I$^2$SB. Finally, we do not include the Schrödinger Bridge methods De Bortoli et al. (2021; 2024); Su et al. (2022); Kim et al. (2023), as they are designed for unpaired data, while we assume paired training data.

**Unification** We were inspired by EDM Karras et al. (2022), which unifies and compares diffusion models within a consistent framework. In a similar direction, Tong et al. (2023) combines the flow-based and Schrödinger Bridge methods for unpaired image translation. For GANs, Lucic et al. (2018) conducted large-scale studies and demonstrated that different architectures achieve comparable results when trained under the same conditions. In image enhancement, Li et al. (2025) provided a comprehensive evaluation of diffusion-based methods using their original backbones. We want to fill this gap by unifying and comparing alternative stochastic processes for image enhancement in the same training setup.

## 5 CONCLUSIONS

In this work, we unify popular stochastic process methods for image enhancement and categorize them as instances of the Ornstein-Uhlenbeck process, diffusion bridge, or standard diffusion process. We separated their definitions from the original schedulers and samplers and summarized them in a structured table. Based on this unification, we implement a library in which each method corresponds directly to the formulas presented in the table, making it easily extensible and straightforward to integrate new techniques. We trained all methods on four image enhancement tasks, including single-image super-resolution, image colorization, low-light image enhancement, and image deraining, ensuring a fair comparison by using the same backbone architecture and identical training settings for each method. Our empirical results show that conditional trajectories offer little to no improvement over standard diffusion. In fact, we observe that plain diffusion often outperforms these techniques, and we suggest that lower temperature may be responsible for the reduced performance.

**Reproducibility statement** We provide the complete code, in the form of a library and framework that was used to train and validate our methods, together with a *README.md* and a requirements file to enable straightforward reproduction of our results.

**Ethics statement** Our work unifies definitions of existing methods and evaluates them on standard benchmarks. We do not introduce new data or methods, therefore, we do not find any ethical concerns associated with this study.

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

# A PROOFS AND METHODS DERIVATIONS

## A.1 FLOW MATCHING DIFFUSION

**Proposition 2.1.** *Let $\mathbf{x}_t$ be a continuous stochastic process that follows given SDE*

$$\mathrm{d}\mathbf{x}_t = -\beta_t \mathbf{x}_t \, \mathrm{d}t + \sqrt{2(1-\phi_t)\beta_t} \, \mathrm{d}\mathbf{w}_t. \tag{5}$$

*If the scheduler is defined as $\beta_t = \frac{1}{1-t}$, then for any $\mathbf{x}_0 \sim p_{data}(\mathbf{x}_0)$ and $\epsilon \sim \mathcal{N}(\mathbf{0}, \boldsymbol{I})$ we have $\mathbf{x}_t = (1-t)\mathbf{x}_0 + t\epsilon$ for $t \in [0,1)$, which corresponds to transitions of Liu et al. (2022).*

*Proof.* To construct the solution, we take the Brownian motion in the interval $[0,1]$ and consider the modified version of the equation 5 in the closed interval $[0,1]$ where $\beta_t$ is replaced by $\tilde{\beta}_t$ defined as follows: $\tilde{\beta}_t = \beta_t$, for $t \in [0, 1-\delta]$ and $\tilde{\beta}_t = \beta_{1-\delta}$, for $t \in [1-\delta, 1]$, for a fixed $\delta \in (0,1)$. The modified equation has a unique strong solution (see Øksendal (2003) Theory 5.2.1 or Karatzas & Shreve (2014) Theory 5.2.5 and 5.5.2.9). If we take $\delta, \delta' \in (0,1)$ and $\delta < \delta'$, due to pathwise uniqueness, almost all trajectories of the corresponding solutions $\mathbf{x}_t^\delta, \mathbf{x}_t^{\delta'}$ agree on $[0, \delta']$. Therefore, $\mathbf{x}_t^\delta$ is an extension of the solution $\mathbf{x}_t^{\delta'}$ of equation 5 from the interval $[0, 1-\delta']$ to $[0, 1-\delta]$. The solution can now be constructed for almost all trajectories by taking $\delta \to 0^+$. Similarly, by truncating the interval $[0,1)$ to $[0, 1-\delta]$ one proves the pathwise uniqueness.

We find that solution of the equation 5 using Integrating Factor method. First, denote that the integrating factor $M$ of equation 5 is equivalent to the definition of $\phi_{s,t}$

$$M = \exp\left(\int_s^t a(z)dz\right) = \phi_{s,t} = \frac{1-t}{1-s}. \tag{17}$$

The mild solution is therefore

$$\mathbf{x}_t = \phi_t \mathbf{x}_0 + \int_0^t \phi_{s,t} g(s) \mathrm{d}\mathbf{w}_s \tag{18}$$

Now, for simplicity, we divide our calculation into expected value $\mathbb{E}[\mathbf{x}_t|\mathbf{x}_0]$ and variance $\mathrm{Var}(\mathbf{x}_t|\mathbf{x}_0)$ of transition densities. Because the Ito integral has zero mean, only the deterministic part of equation 18 contributes to the expected value

$$\mathbb{E}[\mathbf{x}_t|\mathbf{x}_0] = \phi_t \mathbf{x}_0 \tag{19}$$

Similarly, for variance, we consider only the stochastic part of equation 18.

$$\mathrm{Var}(\mathbf{x}_t) = \mathbb{E}\left[\left(\int_0^t \phi_{s,t} g(s) \, \mathrm{d}\mathbf{w}_t\right)^2\right] \tag{20}$$

We can apply Ito isometry to get

$$\mathrm{Var}(\mathbf{x}_t) = \int_0^t \phi_{s,t}^2 g^2(s) ds. \tag{21}$$

Using exponential properties and linearity of Riemann integral we have

$$\phi_{s,t} = \frac{\phi_t}{\phi_s}, \tag{22}$$

we can apply that to equation 21 to obtain

$$\mathrm{Var}(\mathbf{x}_t) = \phi_t^2 \int_0^t \phi_s^{-2} g^2(s) ds. \tag{23}$$

Recall that $g^2(t) = 2(1-\phi_t)\beta_t$ which gives

$$\mathrm{Var}(\mathbf{x}_t) = 2\phi_t^2 \int_0^t \phi_s^{-2} \beta_s (1-\phi_s) ds. \tag{24}$$

We substitute $u = \phi_s^{-1}$, $du = \phi_s^{-1}\beta_s ds$

$$\mathrm{Var}(\mathbf{x}_t) = 2\phi_t^2 \int_1^{\phi_t^{-1}} u - 1 \, du$$

$$= 2\phi_t^2 \left( \frac{\phi_t^{-2}}{2} - \phi_t^{-1} + \frac{1}{2} \right)$$

$$= 1 - 2\phi_t + \phi_t^2$$

$$= (1 - \phi_t)^2 \tag{25}$$

Recall that for our chosen scheduler, we have $\phi_t = 1 - t$, therefore

$$\mathbb{E}[\mathbf{x}_t | \mathbf{x}_0] = (1 - t)\mathbf{x}_0 \tag{26}$$

$$\mathrm{Var}(\mathbf{x}_t) = t^2 \tag{27}$$

and we can express intermediate sample $\mathbf{x}_t$ as

$$\mathbf{x}_t = (1 - t)\mathbf{x}_0 + t\epsilon \tag{28}$$

for some $\mathbf{x}_0 \sim p_{\mathrm{data}}(\mathbf{x}_0)$ and $\epsilon \sim \mathcal{N}(\mathbf{0}, \boldsymbol{I})$. That shows that equation 5 has the same marginals as Liu et al. (2022). $\qquad\square$

## A.2 RESSHIFT

**Proposition 2.2.** *The OU-process with the following stochastic differential equation*

$$\mathrm{d}\mathbf{x}_t = \beta_t(\mathbf{y} - \mathbf{x}_t)\,\mathrm{d}t + \tau\sqrt{\beta_t(2 - \phi_t)}\,\mathrm{d}\mathbf{w}_t \tag{10}$$

*shares the same transition densities as the original ResShift model from equation 9 for $t \in [0, 1]$.*

*Proof.* Based on Theory 5.2.1 from Øksendal (2003), equation 10 has a strong solution and is unique.

Similarly to Proposition 2.1, we can find the solution using the integrating factor method; the subsequent steps are very similar, so we omit some comments.

$$\mathbf{x}_t = \phi_t\mathbf{x}_0 + \mathbf{y}\int_0^t \phi_{s,t}b(s)ds + \int_0^t \phi_{s,t}g(s)\,\mathrm{d}\mathbf{w}_t \tag{29}$$

$$\mathbb{E}[\mathbf{x}_t | \mathbf{x}_0, \mathbf{y}] = \phi_t\mathbf{x}_0 + \mathbf{y}\int_0^t \phi_{s,t}\beta_s ds$$

$$= \phi_t\mathbf{x}_0 + \phi_t\mathbf{y}\int_0^t \phi_s^{-1}\beta_s ds$$

$$= \phi_t\mathbf{x}_0 + \phi_t\mathbf{y}\int_1^{\phi_t^{-1}} dr$$

$$= \phi_t\mathbf{x}_0 + (1 - \phi_t)\mathbf{y} \tag{30}$$

$$\mathrm{Var}(\mathbf{x}_t) = \mathbb{E}\left[ \left( \int_0^t \phi_{s,t}g(s)\,\mathrm{d}\mathbf{w}_t \right)^2 \right]$$

$$= \int_0^t \phi_{s,t}^2 g^2(s)ds$$

$$= \tau^2\phi_t^2 \int_0^t \phi_s^{-2}\beta_s(2 - \phi_s)ds$$

$$= \tau^2\phi_t^2 \int_1^{\phi_t^{-1}} 2r - 1 \, dr$$

$$= \tau^2\phi_t^2 \left( \phi_t^{-2} - \phi_t^{-1} \right)$$

$$= \tau^2(1 - \phi_t) \tag{31}$$

Combining expected value and variance, we get the transition density formula from equation 9. $\quad\square$

### A.3   INDI

**Proposition 2.3.** *The OU-process with the following stochastic differential equation*

$$d\mathbf{x}_t = \beta_t(\mathbf{y} - \mathbf{x}_t)\,dt + \tau\sqrt{2\beta_t(1 - \phi_t)}\,d\mathbf{w}_t. \tag{12}$$

*If the scheduler is defined as $\beta_t = \frac{1}{1-t}$, then for any $\mathbf{x}_0 \sim p_{data}(\mathbf{x}_0)$ and $\epsilon \sim \mathcal{N}(\mathbf{0}, \boldsymbol{I})$ we have the same marginals as equation 11 for $t \in [0, 1)$.*

*Proof.* The conditions are the same as in 2.1.

Similarly to Proposition 2.1, we can find the solution using the integrating factor method; the subsequent steps are very similar, so we omit some comments.

$$\mathbf{x}_t = \phi_t\mathbf{x}_0 + \mathbf{y}\int_0^t \phi_{s,t}\beta_s ds + \int_0^t \tau\phi_{s,t}\sqrt{\beta_s(2 - \phi_s)}d\mathbf{w}_s \tag{32}$$

The expected value is the same as in Proposition 2.2. The variance is a scaled version of the variance from the Proposition 2.1. Therefore, we have

$$\mathbb{E}[\mathbf{x}_t|\mathbf{x}_0, \mathbf{y}] = \phi_t\mathbf{x}_0 + (1 - \phi_t)\mathbf{y} \tag{33}$$

$$\mathrm{Var}(\mathbf{x}_t) = \tau^2(1 - \phi_t)^2 \tag{34}$$

If we set $\phi_t = 1 - t$ then we get

$$p_t(\mathbf{x}_t|\mathbf{x}_0, \mathbf{y}) = \mathcal{N}(\mathbf{x}_t; (1 - t)\mathbf{x}_0 + t\mathbf{y}, \tau^2 t^2 \boldsymbol{I}), \tag{35}$$

which is equivalent to equation 11. $\qquad\square$

### A.4   I$^2$SB

**Remark 2.2.** *Deriving the bridge process from Liu et al. (2023) Theory 3.1 with $\mathbf{f}(\mathbf{x}_t, t) = \mathbf{0}$ and $g(t) = \sqrt{\beta_t}$ leads to the same result as applying h-transform Doob & Doob (1984) to the exact same SDE $d\mathbf{x}_t = \sqrt{\beta_t}\,d\mathbf{w}_t$. This gives the definition of DDBM-VE Zhou et al. (2023b).*

We consider two SDEs where, respectively, time goes forward and backward

$$d\mathbf{x}_t = [\mathbf{f}(\mathbf{x}_t, t) + g^2(t)\nabla_{\mathbf{x}_t}\log\Psi(\mathbf{x}_t, t)]\,dt + g(t)\,d\mathbf{w}_t, \tag{36}$$

$$d\mathbf{x}_t = [\mathbf{f}(\mathbf{x}_t, t) - g^2(t)\nabla_{\mathbf{x}_t}\log\hat{\Psi}(\mathbf{x}_t, t)]\,dt + g(t)\,d\bar{\mathbf{w}}_t, \tag{37}$$

$\mathbf{x}_0 \sim p_{data}(\mathbf{x}_0)$, $\mathbf{x}_1 \sim p_{low}(\mathbf{x}_1)$, and the functions $\Psi$ and $\hat{\Psi}$ are the solution to the following coupled PDEs[5]

$$\begin{cases} \dfrac{\partial\Psi}{\partial t} = -\nabla_{\mathbf{x}}\Psi^{\mathsf{T}}\mathbf{f} - \dfrac{1}{2}\mathrm{Tr}(g^2\nabla_{\mathbf{x}}^2\Psi), \\[2mm] \dfrac{\partial\hat{\Psi}}{\partial t} = -\nabla_{\mathbf{x}}\cdot(\hat{\Psi}\mathbf{f}) + \dfrac{1}{2}\mathrm{Tr}(g^2\nabla_{\mathbf{x}}^2\hat{\Psi}), \end{cases} \tag{38}$$

$$s.t. \quad \Psi(\mathbf{x}, 0)\hat{\Psi}(\mathbf{x}, 0) = p_{data}(\mathbf{x}), \quad \Psi(\mathbf{x}, 1)\hat{\Psi}(\mathbf{x}, 1) = p_{low}(\mathbf{x}). \tag{39}$$

From Theory 3.1 from Liu et al. (2023) when the above system holds, we can consider $\hat{\Psi}(\cdot, 0), \Psi(\cdot, 1)$ as the boundary distributions and $\nabla_{\mathbf{x}_t}\log\hat{\Psi}(\mathbf{x}_t, t)$ and $\nabla_{\mathbf{x}_t}\log\Psi(\mathbf{x}_t, t)$ are the score functions for the following SDEs, respectively

$$d\mathbf{x}_t = \mathbf{f}(\mathbf{x}_t, t)\,dt + g(t)\,d\mathbf{w}_t, \quad \mathbf{x}_0 \sim \hat{\Psi}(\mathbf{x}_0, 0) \tag{40}$$

$$d\mathbf{x}_t = \mathbf{f}(\mathbf{x}_t, t)\,dt + g(t)\,d\bar{\mathbf{w}}_t, \quad \mathbf{x}_1 \sim \Psi(\mathbf{x}_1, 1), \tag{41}$$

with the same $\mathbf{f}$ and $g$ as in equation 36 and equation 37. For simplicity, we specify the drift $\mathbf{f}(\mathbf{x}_t, t) = \mathbf{0}$ and the diffusion coefficient $g(t) = \sqrt{\beta_t}$ because that was the practical design choice

---

[5]Following Chen et al. (2021), for brevity we denote $\Psi \equiv \Psi(\mathbf{x}, t), \mathbf{f} \equiv \mathbf{f}(\mathbf{x}, t)$, and $g \equiv g(t)$.

made in Liu et al. (2023). With that we can easily calculate the transition kernel for equation 41 remembering that time goes backwards

$$p_t(\mathbf{x}_t|\mathbf{x_1}) = \mathcal{N}(\mathbf{x}_t; \mathbf{x_1}, \int_t^1 \beta_s ds \boldsymbol{I}). \tag{42}$$

To remain consistent with our notation, we denote $\alpha_{t,1} = \int_t^1 \beta_s ds$. With that, we can define the score of equation 41 as $\nabla_{x_t} \log \Psi(\mathbf{x}_t, t) = \frac{\mathbf{x_1} - \mathbf{x}_t}{\alpha_{t,1}}$. Finally, we substitute the definitions of $\mathbf{f}$, $g$, and $\nabla_{x_t} \log \Psi(\mathbf{x}_t, t)$ into equation 36 which yields

$$d\mathbf{x}_t = \frac{\beta_t}{\alpha_{t,1}}(\mathbf{x_1} - \mathbf{x}_t) \, dt + \sqrt{\beta_t} \, d\mathbf{w}_t. \tag{43}$$

To further unify the notation, we denote $\mathbf{y} \equiv \mathbf{x_1}$, which is true for all the models based on bridges that we consider

$$d\mathbf{x}_t = \frac{\beta_t}{\alpha_{t,1}}(\mathbf{y} - \mathbf{x}_t) \, dt + \sqrt{\beta_t} \, d\mathbf{w}_t. \tag{44}$$

This corresponds to the h-transform Doob & Doob (1984) applied to variance exploding diffusion process Zhou et al. (2023b); Li et al. (2023).

## B  UNIFIED SCHEDULERS

In this section, we show all the schedulers that were used in the methods we covered. Each scheduler is defined for $t \in [0, 1]$.

We provide the exact definitions for $\beta_t$ and $\alpha_{s,t} = \int_s^t \beta_z dz$. The $\phi_{s,t} = \exp(-\alpha_{s,t})$ can be easily computed for any scheduler.

**Linear**   with two parameters $\beta_{min}$, and $\beta_{max}$ that denote the value of $\beta_t$ at time 0, and 1.

$$\beta_t = (\beta_{max} - \beta_{min})t + \beta_{min} \tag{45}$$

$$\alpha_{s,t} = \frac{1}{2}(\beta_{max} - \beta_{min})(t^2 - s^2) + \beta_{min}(t - s) \tag{46}$$

**Cosine**   was used among several papers with two parameters $\epsilon$ and $\delta$ that are used to ensure numerical stability. For all methods, we have

$$\epsilon = 0.008, \delta = 0.005.$$

For clarity, we shall define a few intermediate functions

$$g(t) = \cos\left(\frac{t + \epsilon}{1 + \epsilon} \cdot \frac{\pi}{2}\right)^2 \tag{47}$$

$$h(t) = \sin\left(\frac{t + \epsilon}{1 + \epsilon}\pi\right) \tag{48}$$

$$f(t) = 1 - \frac{g(t)}{g(0)} \tag{49}$$

$$F(s, t) = \int_s^t f(z)dz = t - s + \frac{(s - t)\pi + (1 + \epsilon)(h(s) - h(t))}{2\pi g(0)} \tag{50}$$

Now, we can easily define the scheduler

$$\beta_t = -\log(\delta)\frac{f(t)}{F(0, 1)} \tag{51}$$

$$\alpha_{s,t} = -\log(\delta)\frac{F(s, t)}{F(0, 1)} \tag{52}$$

**Exponential** with three parameters $\eta_{min}$, $\eta_{max}$, and $p$, where $\eta_{min}$ and $\eta_{max}$ are responsible for a value range of $\phi$, and $p$ influence how fast $\phi$ goes to $\eta_{max}$.

$$\beta_t = 2p \log\left(\frac{\eta_{max}}{\eta_{min}}\right) t^{p-1} \eta_{min}^2 \left(\frac{\eta_{max}}{\eta_{min}}\right)^{2t^p} \phi_t \tag{53}$$

$$\alpha_{s,t} = \log\left(\frac{1 - \left(\eta_{min}^2 \left(\frac{\eta_{max}}{\eta_{min}}\right)^{2s^p}\right)}{1 - \left(\eta_{min}^2 \left(\frac{\eta_{max}}{\eta_{min}}\right)^{2t^p}\right)}\right) \tag{54}$$

**Inversed** is defined so that $\phi_t = 1 - t$ and it is a useful scheduler for methods based on linear interpolation such as Flow Matching or InDI.

$$\beta_t = \frac{1}{1-t} \tag{55}$$

$$\alpha_{s,t} = \log\left(\frac{1-s}{1-t}\right) \tag{56}$$

**Quadratic Symmetric** is used when we want the transition variance of a diffusion bridge to be symmetric at $t \in [0,1]$.

$$\beta_t = \left(\left(\sqrt{\beta_{max}} - \sqrt{\beta_{min}}\right)\left(\frac{1}{2} - \left|\frac{1}{2} - t\right|\right) + \sqrt{\beta_{min}}\right)^2 \tag{57}$$

$$\tag{58}$$

We consider two scenarios $t \leq 0.5$ and $t > 0.5$.

$$f(t)^{(1)} = \left(\sqrt{\beta_{max}} - \sqrt{\beta_{min}}\right)^2 \frac{t^3}{3} + \left(\sqrt{\beta_{max}} - \sqrt{\beta_{min}}\right)\sqrt{\beta_{min}}t^2 + \beta_{min}t \tag{59}$$

$$f(t)^{(2)} = \left(\sqrt{\beta_{max}} - \sqrt{\beta_{min}}\right)^2 \cdot \left(t - t^2 + \frac{t^3}{3}\right) + \tag{60}$$

$$\left(\sqrt{\beta_{max}} - \sqrt{\beta_{min}}\right)\sqrt{\beta_{min}} \cdot (2t - t^2) + \beta_{min}t \tag{61}$$

$$\alpha_t = \begin{cases} f(t)^{(1)}, & t \leq \frac{1}{2} \\ f(t)^{(2)} - f(t)^{(2)} + f(t)^{(1)}, & t > \frac{1}{2} \end{cases} \tag{62}$$

$$\alpha_{s,t} = \alpha_t - \alpha_s \tag{63}$$

## C  LIBRARY

*Ito Vision* is a Python library, built on PyTorch, that implements 11 considered methods, together with multiple schedulers, samplers, network parametrizations and discretization techniques. Collectively, these components support the use of all methods presented in this paper and enable straightforward implementation of novel approaches. The library can be installed via:

```
$ pip install <path/to/supplement/ito_vision>
```

The codebase is fully type-hinted and designed for consistency through the use of abstract classes for each module. For example, every method inherits from the `IterativeRefinementMethod` abstract class, which specifies required functions that correspond to the definitions from Table 1, bridging the gap between the mathematical formulations and the code, and enabling easy implementation of new stochastic processes.

Further details and a code example are provided in the library's `README.md`, included in the supplementary material.

# D IMPLEMENTATION DETAILS

Table 3: Original choices of used scheduler, sampler, and other method-specific parameters.

| Method | Scheduler $\beta_t$ | Sampler | Parametrization | Loss | Parameters |
|---|---|---|---|---|---|
| DM-VP† | linear | Ancestral sampling | $\epsilon_t$ | $l_2$ norm | - |
| FM | inversed | Euler-ODE | $\mathbf{x}_0 - \mathbf{y}$ | $l_2$ norm | - |
| IR-SDE | cosine | Euler-Maruyama | $\epsilon_t$ | $l_2$ norm | $\tau = 0.20$ |
| ResShift | exponential | Ancestral sampling | $\mathbf{x}_0$ | $l_2$ norm | $\tau = 2.00$ |
| InDI | inversed | Euler-ODE | $\mathbf{x}_0$ | $l_1$ norm | $\tau = 0.06$ |
| BBDM | constant | Ancestral sampling | $\epsilon_t$ | $l_2$ norm | - |
| DDBM-VE | linear | 2nd Heun & Langevin-Heun | karras | $l_2$ norm | - |
| DDBM-VP | linear | 2nd Heun & Langevin-Heun | karras | $l_2$ norm | - |
| I$^2$SB | quadratic-symmetric | Ancestral sampling | $\epsilon_t$ | $l_2$ norm | - |
| GOUB | cosine | Euler-ODE & mean-ODE | $\epsilon_t$ | $l_1$ norm | $\tau = 0.34$ |
| UniDB-GOUB | cosine | Euler-ODE & mean-ODE | $\epsilon_t$ | $l_1$ norm | $\tau = 0.34, \gamma = 1e4$ |

† For diffusion we consider DDPM Ho et al. (2020) implementation.

Table 4: Detailed parameters choices of training procedure for each image enhancement task.

| | Super-Resolution | Colorization | Low-light Enhancement | Deraining |
|---|---|---|---|---|
| Dataset | FFHQ | ImageNet | LOL | Rain100H |
| Iterations | 400k | 400k | 150k | 50k |
| Unet parameters | 119M | 119M | 31.3M | 31.3M |
| Channels | 128 | 128 | 64 | 64 |
| Depth | 2 | 2 | 2 | 2 |
| Channel multipliers | 1,1,2,2,4,4 | 1,1,2,2,4,4 | 1,1,2,2,4,4 | 1,1,2,2,4,4 |
| Attention head dimension | 64 | 64 | 64 | 64 |
| Latent | ✗ | ✓ | ✗ | ✗ |
| Learning rate† | 1e-4 → 1e-7 | 1e-4 → 1e-7 | 1e-4 → 1e-7 | 1e-4 → 1e-7 |
| Random crop size | $256 \times 256$ | $384 \times 384$ | $320 \times 320$ | $320 \times 320$ |
| Batch size | 64 | 256 | 80 | 80 |

† Decreasing learning rate with respect to cosine annealing LR scheduler.

Below we provide detailed description of the experimental setups for each task. We evaluate the methods using Peak Signal-to-Noise Ratio (PSNR), Structural Similarity Index Measure (SSIM), Learned Perceptual Image Patch Similarity (LPIPS) Zhang et al. (2018), and Natural Image Quality Evaluator (NIQE) Mittal et al. (2012). For image super-resolution and colorization, we also report the Fréchet Inception Distance (FID) Heusel et al. (2017), as the test datasets are large enough for this measure. We use LPIPS as the main metric to evaluate model quality because, unlike PSNR and SSIM, it aligns with human perception, and unlike NIQE and FID, it incorporates ground truth, which is important for preserving details.

**Image super-resolution** For this task, we used the FFHQ dataset Karras et al. (2019) and performed x8 single image super-resolution $64 \times 64 \rightarrow 512 \times 512$. In addition to reducing resolution, we also applied JPEG compression to $10\%$ quality and then upscaled back to $512 \times 512$. Both resizing used bicubic interpolation with anti-aliasing. Each model was trained in $256 \times 256$ crops with batch size of 64. We split the data set that contains 70k images into train-val-test splits with $0.98 : 0.002 : 0.018$ proportions. For our backbone, we used UNet Ronneberger et al. (2015) with 119M parameters from the diffuser library von Platen et al. (2022) with the same channel dimensionality as in Dhariwal & Nichol (2021) (see their Appendix I, Table 12 ImageNet $256 \times 256$). Detailed architectures are shown in Table 4. Each method was trained for 400k iterations. The final model was selected based on the LPIPS metric Zhang et al. (2018) on the validation split, measured each 10k iterations.

**Low-light image enhancement** We used the LOL dataset Wei et al. (2018), as it contains a relatively extensive collection of 500 real-world dark and light pairs of the same scene. We used original train and test splits, and we moved 5 pairs from training to validation split as it was not provided. We trained the models on $320 \times 320$ crops with batch size of 80. The model had 31.3M parameters and its architecture was the same as in the super-resolution task, with reduced channel dimensions. Each method was trained for 150k iterations with the validation each 2.5k iterations. Following most of the state-of-the-art methods in this field Zhang et al. (2019); Cai et al. (2023); Zhou et al. (2023a) we involve conditioning on ground-truth average lightness. We injected it through the attention mechanism and adjusted the value channel for each estimation of ground truth $\hat{\mathbf{x}}_{0|t}$ to the given lightness through the HSV color space.

**Image colorization** Here, we utilized the ImageNet dataset Russakovsky et al. (2015). It contains a wide range of real world objects; therefore, it fits this task perfectly as the diversity of objects and their hues makes it well-suited for assessing generalization of validated methods. For training, we used all images from the train split, with width and height between 256 and 768. We treat low-quality input $\mathbf{y}$ as the grayscale version of ground truth with three channels to maintain the same dimensionality. We used pretrained VAE from Stable Diffusion 2.1 Rombach et al. (2022) originally used for $\times 4$ super-resolution. We use the same architecture of the UNet as in the image super-resolution task. We train it for 400k iteration with $384 \times 384$ random crops and a batch size of 256. We evaluated FID on 50k images from the test split.

**Image deraining** For our last task we used the Rain1400 dataset Fu et al. (2017) with prepared image pairs with synthetic rain applied to low-quality images $\mathbf{y}$. Similarly to other experiments, we report PSNR and SSIM metrics in RGB space.

# E    THE USE OF LARGE LANGUAGE MODELS

We used LLMs to find better synonyms and correct grammar in our text to improve its readability. We read, analyzed, and modified each LLM suggestion, if needed, to ensure that there were no hallucinations that could lead to misinformation.

# F    ADDITIONAL RESULTS

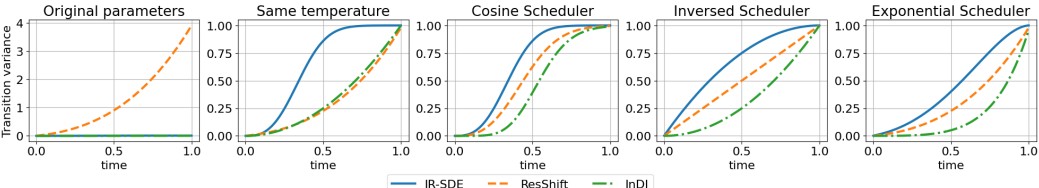

Figure 3: Transition variance $\mathrm{Var}(\mathbf{x}_t|\mathbf{x}_0)$ for all considered Ornstein-Uhlenbeck processes. Left: Original temperatures $\tau$ and schedulers $\beta_t$. Next: standarized $\tau$ with original schedulers. Middle to right: standarized $\tau$ and specific schedulers. We can see that ResShift has by far the highest temperature but IR-SDE reaches maximum level of noise faster than other methods. InDI has the lowest temperature and adds the noise with the slowest pace.

Table 5: Image super-resolution on FFHQ dataset using ancestral sampling with Network forward evaluations (NFE) = 5, 35, and 100. The best values are **bolded** and second to best are underscored.

| Model | NFE = 5 | | | | | NFE = 35 | | | | | NFE = 100 | | | | |
|---|---|---|---|---|---|---|---|---|---|---|---|---|---|---|---|
| | PSNR | SSIM | LPIPS | NIQE | FID | PSNR | SSIM | LPIPS | NIQE | FID | PSNR | SSIM | LPIPS | NIQE | FID |
| DDPM | **29.04** | **0.7825** | 0.229 | 8.728 | 17.857 | **27.86** | **0.7481** | **0.181** | 7.979 | 17.525 | **27.23** | 0.7217 | **0.164** | 7.652 | 17.760 |
| FM | 28.72 | 0.7756 | 0.241 | 8.818 | 17.565 | 27.15 | 0.7295 | 0.183 | 7.839 | 17.745 | 26.48 | 0.6989 | 0.172 | 7.561 | 17.843 |
| IR-SDE | 28.64 | 0.7712 | 0.249 | 8.974 | 18.230 | 26.97 | 0.7179 | 0.184 | **7.805** | 18.302 | 26.17 | 0.6881 | 0.189 | 7.631 | 18.473 |
| ResShift | 28.84 | 0.7781 | **0.222** | **8.679** | **17.517** | 27.66 | 0.7461 | 0.184 | 7.965 | **17.422** | 27.23 | **0.7304** | 0.175 | 7.722 | **17.549** |
| InDI | 28.56 | 0.7657 | 0.319 | 9.759 | 19.820 | 27.21 | 0.7125 | 0.199 | 7.984 | 18.861 | 26.40 | 0.6801 | 0.195 | 7.643 | 18.893 |
| BBDM | 28.06 | 0.7529 | 0.278 | 9.157 | 19.533 | 27.14 | 0.7282 | 0.252 | 8.550 | 19.435 | 26.83 | 0.7176 | 0.245 | 8.356 | 19.513 |
| DDBM-VE | 28.55 | 0.7683 | 0.237 | 8.791 | 17.830 | 27.11 | 0.7239 | 0.186 | 7.813 | 17.749 | 26.60 | 0.7021 | 0.178 | 7.556 | 18.079 |
| DDBM-VP | 28.75 | 0.7768 | 0.232 | 8.750 | 17.598 | 27.46 | 0.7410 | 0.190 | 7.966 | 17.584 | 26.98 | 0.7217 | 0.176 | 7.663 | 17.581 |
| I$^2$SB | 28.67 | 0.7708 | 0.245 | 8.960 | 17.938 | 27.46 | 0.7333 | 0.184 | 7.873 | 17.786 | 27.03 | 0.7173 | 0.174 | 7.647 | 17.789 |
| GOUB | 28.52 | 0.7680 | 0.270 | 9.197 | 18.688 | 27.24 | 0.7347 | 0.224 | 8.246 | 18.819 | 26.40 | 0.7136 | 0.213 | 7.922 | 19.076 |
| UniDB | 28.91 | 0.7789 | 0.247 | 8.981 | 17.602 | 27.59 | 0.7387 | **0.181** | 7.883 | 17.589 | 26.28 | 0.7023 | 0.177 | **7.514** | 18.082 |

Table 6: Low-light image enhancement on LOL dataset using ancestral sampling with Network forward evaluations (NFE) = 5, 35, and 100. The best values are **bolded** and second to best are underscored.

| Model | NFE = 5 | | | | NFE = 35 | | | | NFE = 100 | | | |
|---|---|---|---|---|---|---|---|---|---|---|---|---|
| | PSNR | SSIM | LPIPS | NIQE | PSNR | SSIM | LPIPS | NIQE | PSNR | SSIM | LPIPS | NIQE |
| DM-VP | 23.01 | 0.6855 | **0.185** | **5.517** | 23.01 | 0.6846 | **0.186** | 5.506 | 23.01 | 0.6842 | **0.186** | 5.514 |
| FM | 22.78 | 0.6848 | 0.202 | 5.677 | 22.57 | 0.6781 | 0.209 | 5.646 | 22.47 | 0.6756 | 0.215 | 5.668 |
| IR-SDE | **23.43** | 0.6960 | 0.208 | 5.745 | 23.07 | 0.6785 | 0.237 | 5.549 | 23.00 | 0.6736 | 0.248 | 5.536 |
| ResShift | 23.13 | 0.6926 | 0.194 | 5.756 | 23.15 | **0.6928** | 0.195 | 5.764 | **23.15** | **0.6928** | 0.195 | 5.753 |
| InDI | 21.26 | 0.6409 | 0.278 | 5.530 | 21.48 | 0.6327 | 0.323 | 5.439 | 21.35 | 0.6250 | 0.334 | 5.487 |
| BBDM | 22.84 | 0.6872 | 0.190 | 5.794 | 22.84 | 0.6869 | 0.191 | 5.788 | 22.84 | 0.6866 | 0.191 | 5.787 |
| DDBM-VE | **23.43** | 0.6933 | 0.201 | 5.555 | **23.18** | 0.6795 | 0.224 | **5.403** | 23.10 | 0.6752 | 0.234 | **5.383** |
| DDBM-VP | 23.16 | 0.6815 | 0.201 | 5.550 | 22.55 | 0.6587 | 0.231 | 5.443 | 22.38 | 0.6520 | 0.243 | 5.447 |
| I$^2$SB | 22.01 | 0.6617 | 0.238 | 5.834 | 21.86 | 0.6552 | 0.245 | 5.769 | 21.86 | 0.6547 | 0.248 | 5.751 |
| GOUB | 23.11 | 0.6919 | 0.207 | 5.599 | 22.57 | 0.6690 | 0.233 | 5.513 | 22.38 | 0.6600 | 0.243 | 5.531 |
| UniDB | 23.36 | **0.6989** | 0.197 | 5.625 | 22.80 | 0.6793 | 0.223 | 5.609 | 22.56 | 0.6715 | 0.235 | 5.592 |

Table 7: Image colorization on ImageNet dataset using ancestral sampling with Network forward evaluations (NFE) = 5, 35, and 100. The best values are **bolded** and second to best are underscored. For this task, we also provide FID metric evaluated on 50k samples.

| Model | NFE = 5 | | | | | NFE = 35 | | | | | NFE = 100 | | | | |
|---|---|---|---|---|---|---|---|---|---|---|---|---|---|---|---|
| | PSNR | SSIM | LPIPS | NIQE | FID | PSNR | SSIM | LPIPS | NIQE | FID | PSNR | SSIM | LPIPS | NIQE | FID |
| DM-VP | 22.09 | 0.6969 | 0.186 | 4.772 | 3.859 | 21.73 | 0.6852 | 0.191 | 4.714 | 3.994 | 21.57 | 0.6797 | 0.194 | 4.690 | 4.085 |
| FM | 22.32 | 0.7033 | 0.183 | 4.795 | 3.452 | 21.64 | 0.6850 | 0.191 | 4.762 | 3.500 | 21.31 | 0.6750 | 0.196 | 4.740 | 3.575 |
| IR-SDE | 22.35 | 0.7035 | 0.186 | 4.803 | 3.597 | 22.02 | 0.6905 | 0.187 | 4.730 | 3.751 | 21.82 | 0.6823 | 0.190 | 4.693 | 3.847 |
| ResShift | 22.39 | 0.7058 | 0.183 | 4.822 | 3.473 | 21.82 | 0.6894 | 0.189 | 4.808 | 3.481 | 21.55 | 0.6809 | 0.194 | 4.801 | 3.487 |
| InDI | 22.39 | 0.7044 | 0.192 | 4.811 | 4.449 | 22.28 | 0.6992 | 0.191 | 4.747 | 4.416 | 22.06 | 0.6914 | 0.190 | 4.715 | 4.444 |
| BBDM | 22.44 | 0.7063 | **0.179** | 4.823 | **3.306** | 22.17 | 0.6977 | **0.182** | 4.789 | **3.323** | 21.98 | 0.6916 | **0.185** | 4.780 | **3.324** |
| DDBM-VE | **22.57** | **0.7064** | 0.180 | 4.729 | 3.842 | 22.36 | 0.7003 | **0.181** | 4.737 | 3.712 | 22.17 | 0.6940 | **0.183** | 4.692 | 3.732 |
| DDBM-VP | 22.50 | 0.7052 | 0.182 | **4.671** | 3.487 | 22.23 | 0.6969 | 0.184 | **4.664** | 3.536 | 22.02 | 0.6897 | 0.187 | 4.631 | 3.543 |
| I$^2$SB | 22.52 | 0.7058 | 0.183 | 4.838 | 4.402 | **22.39** | **0.7009** | 0.182 | 4.846 | 4.259 | **22.28** | **0.6972** | **0.183** | 4.830 | 4.208 |
| GOUB | 22.17 | 0.6993 | 0.188 | 4.770 | 3.629 | 21.59 | 0.6808 | 0.193 | 4.691 | 3.799 | 18.54 | 0.5938 | 0.270 | **4.598** | 4.256 |
| UniDB | 22.20 | 0.6999 | 0.186 | 4.787 | 3.637 | 21.69 | 0.6835 | 0.190 | 4.727 | 3.806 | 19.32 | 0.6153 | 0.248 | 4.654 | 4.130 |

Table 8: Image deraining on Rain1400 dataset using ancestral sampling with Network forward evaluations (NFE) = 5, 35, and 100. The best values are **bolded** and second to best are underscored.

| Model | NFE = 5 | | | | NFE = 35 | | | | NFE = 100 | | | |
|---|---|---|---|---|---|---|---|---|---|---|---|---|
| | PSNR | SSIM | LPIPS | NIQE | PSNR | SSIM | LPIPS | NIQE | PSNR | SSIM | LPIPS | NIQE |
| DM-VP | 29.92 | 0.8703 | 0.052 | 4.356 | 29.02 | 0.8528 | 0.055 | 4.243 | 28.56 | 0.8450 | 0.058 | 4.239 |
| FM | 29.98 | 0.8702 | 0.050 | 4.290 | 28.89 | 0.8444 | 0.057 | 4.154 | 28.46 | 0.8352 | 0.062 | 4.153 |
| IR-SDE | 29.54 | 0.8627 | 0.048 | 4.217 | 28.15 | 0.8269 | 0.061 | 4.104 | 27.52 | 0.8115 | 0.070 | 4.112 |
| ResShift | 29.69 | 0.8670 | 0.050 | 4.283 | 29.17 | 0.8562 | 0.052 | 4.217 | 28.99 | 0.8527 | 0.052 | 4.197 |
| InDI | 29.93 | 0.8633 | 0.050 | **4.194** | 28.17 | 0.8159 | 0.069 | **4.081** | 27.60 | 0.7995 | 0.079 | **4.087** |
| BBDM | **30.37** | **0.8784** | 0.048 | 4.366 | **30.10** | **0.8731** | **0.049** | 4.349 | **30.04** | **0.8723** | **0.049** | 4.348 |
| DDBM-VE | 29.55 | 0.8595 | 0.050 | 4.293 | 28.65 | 0.8372 | 0.054 | 4.133 | 28.37 | 0.8276 | 0.058 | 4.105 |
| DDBM-VP | 29.82 | 0.8632 | 0.049 | 4.298 | 28.94 | 0.8422 | 0.052 | 4.162 | 28.60 | 0.8315 | 0.056 | 4.109 |
| I$^2$SB | 30.07 | 0.8715 | **0.047** | 4.264 | 29.29 | 0.8542 | 0.050 | 4.165 | 29.14 | 0.8508 | 0.051 | 4.158 |
| GOUB | 29.79 | 0.8673 | 0.050 | 4.277 | 28.38 | 0.8313 | 0.061 | 4.102 | 27.73 | 0.8140 | 0.072 | 4.097 |
| UniDB | 29.38 | 0.8606 | 0.049 | 4.239 | 28.02 | 0.8237 | 0.060 | 4.092 | 27.29 | 0.8031 | 0.073 | 4.116 |

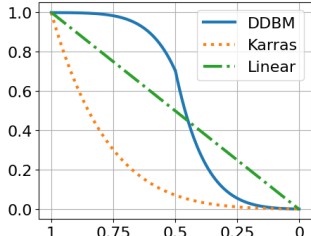

Figure 4: Visualization of different discretization techniques. Karras discretization spends most of the time at low values of $t$, while DDBM places emphasis on both the beginning and the end of the trajectory.

Table 9: Different samplers used in image super-resolution for diffusion and Ornstein-Uhlenbeck processes. We measure their effectiveness using PSNR / SSIM / LPIPS metrics for Network Forward Evaluations (NFE) = 5, 35, and 100. Best results across samplers are **bolded** and second best are underscored. Ancestral sampling and Euler ODE achieve the best LPIPS results. Mean-reverting ODE yields the highest PSNR and SSIM but low LPIPS, suggesting possible oversmoothing. Our Exponential Integrator (EI-ODE) relies on numerical methods, which perform worse for flow matching and InDI due to their stiff dynamics. Surprisingly, Heun performs worst among the second-order Runge-Kutta methods.

| NFE | Sampler | DM-VP | FM | IR-SDE | ResShift | InDI |
|---|---|---|---|---|---|---|
| 5 | Euler ODE | 25.2 / 0.51 / 0.40 | 28.4 / 0.77 / **0.22** | 26.3 / 0.58 / 0.35 | 28.2 / 0.76 / **0.20** | 28.5 / 0.76 / **0.29** |
| | Euler SDE | 17.5 / 0.18 / 1.00 | 8.9 / 0.01 / 0.84 | 18.7 / 0.13 / 0.82 | 18.5 / 0.15 / 1.07 | 7.6 / 0.00 / 1.49 |
| | Ancestral† | 29.0 / 0.78 / **0.23** | 28.7 / 0.78 / 0.24 | 28.6 / 0.77 / **0.25** | 28.8 / 0.78 / 0.22 | 28.6 / **0.77** / 0.32 |
| | EI-ODE | 14.9 / 0.11 / 0.97 | 9.5 / 0.01 / 0.84 | 14.5 / 0.13 / 0.75 | 18.1 / 0.18 / 0.94 | 3.0 / -0.39 / 0.82 |
| | Mean-ODE†† | 21.4 / 0.29 / 0.79 | 8.9 / 0.01 / 0.84 | 21.7 / 0.23 / 0.60 | 20.0 / 0.19 / 1.00 | 7.6 / 0.00 / 1.49 |
| | Langevin-Heun††† | **29.5** / **0.79** / 0.28 | **29.3** / **0.79** / 0.29 | **29.2** / **0.79** / 0.30 | **29.5** / **0.79** / 0.28 | **28.7** / **0.77** / 0.34 |
| | 2nd Heun | 15.8 / 0.13 / 0.92 | 13.0 / 0.30 / 0.61 | 9.0 / 0.01 / 0.94 | 12.5 / 0.03 / 0.87 | 12.9 / 0.33 / 0.59 |
| | 2nd Midpoint | 20.7 / 0.34 / 0.66 | 28.1 / 0.76 / 0.23 | 25.9 / 0.62 / **0.25** | 24.6 / 0.47 / 0.43 | 28.5 / 0.76 / **0.30** |
| | 2nd Ralston | 20.4 / 0.60 / 0.37 | 27.9 / 0.75 / **0.22** | 22.7 / 0.27 / 0.57 | 22.5 / 0.31 / 0.65 | 28.6 / **0.77** / 0.31 |
| 35 | Euler ODE | 27.0 / 0.71 / **0.16** | 26.7 / 0.71 / **0.17** | 26.4 / 0.68 / 0.19 | 27.2 / 0.73 / **0.18** | 26.8 / 0.69 / **0.19** |
| | Euler SDE | 27.5 / 0.70 / 0.25 | 25.5 / 0.55 / 0.29 | 25.7 / 0.62 / 0.25 | 27.5 / 0.74 / 0.19 | 16.7 / 0.15 / 0.84 |
| | Ancestral | 27.9 / 0.75 / 0.18 | 27.2 / 0.73 / 0.18 | 27.0 / 0.72 / **0.18** | 27.7 / 0.75 / **0.18** | **27.2** / **0.71** / 0.20 |
| | EI-ODE | 27.3 / 0.73 / 0.25 | 13.1 / 0.17 / 0.75 | 25.6 / 0.70 / 0.21 | 27.3 / 0.75 / 0.22 | 11.5 / 0.44 / 0.64 |
| | Mean-ODE | **28.2** / **0.78** / 0.32 | **28.2** / **0.78** / 0.32 | **27.9** / **0.77** / 0.32 | **28.2** / **0.78** / 0.30 | 25.8 / 0.67 / 0.45 |
| | Langevin-Heun | 25.2 / 0.58 / 0.38 | 12.6 / 0.19 / 0.91 | 22.8 / 0.48 / 0.35 | 19.8 / 0.24 / 0.74 | 5.8 / -0.00 / 0.92 |
| | 2nd Heun | 25.6 / 0.63 / 0.28 | 23.2 / 0.58 / 0.27 | 23.8 / 0.57 / 0.26 | 25.1 / 0.44 / 0.62 | 24.8 / 0.65 / 0.23 |
| | 2nd Midpoint | 26.9 / 0.70 / 0.17 | 26.3 / 0.69 / 0.18 | 26.1 / 0.67 / 0.19 | 26.9 / 0.72 / **0.18** | 26.1 / 0.66 / 0.20 |
| | 2nd Ralston | 26.8 / 0.70 / 0.17 | 26.2 / 0.68 / 0.18 | 26.1 / 0.67 / 0.19 | 26.9 / 0.72 / **0.18** | 26.4 / 0.67 / 0.20 |
| 100 | Euler ODE | 26.8 / 0.69 / 0.17 | 26.4 / 0.69 / **0.17** | 26.1 / 0.69 / **0.19** | 27.0 / 0.72 / 0.18 | 26.2 / 0.67 / 0.20 |
| | Euler SDE | 27.2 / 0.73 / 0.18 | 25.9 / 0.65 / 0.20 | 25.6 / 0.65 / 0.21 | 27.2 / 0.73 / 0.18 | 21.6 / 0.50 / 0.39 |
| | Ancestral | 27.2 / 0.72 / **0.16** | 26.5 / 0.70 / **0.17** | 26.2 / 0.69 / **0.19** | 27.2 / 0.73 / **0.17** | 26.4 / 0.68 / **0.19** |
| | EI-ODE | 26.9 / 0.72 / 0.18 | 17.0 / 0.21 / 0.69 | 25.8 / 0.68 / **0.19** | 27.0 / 0.72 / **0.17** | 13.1 / 0.48 / 0.56 |
| | Mean-ODE | **28.2** / **0.78** / 0.32 | **28.2** / **0.78** / 0.32 | **27.6** / **0.77** / 0.33 | **28.1** / **0.78** / 0.30 | **27.3** / **0.75** / 0.35 |
| | Langevin-Heun | 26.4 / 0.67 / 0.19 | 13.6 / 0.17 / 0.70 | 25.1 / 0.63 / 0.23 | 27.1 / 0.72 / 0.25 | 8.6 / 0.04 / 0.99 |
| | 2nd Heun | 26.3 / 0.67 / 0.19 | 25.3 / 0.65 / 0.21 | 25.3 / 0.65 / 0.21 | 27.3 / 0.73 / 0.24 | 25.4 / 0.65 / 0.21 |
| | 2nd Midpoint | 26.6 / 0.68 / 0.18 | 26.2 / 0.67 / 0.18 | 25.9 / 0.67 / 0.20 | 26.8 / 0.71 / 0.18 | 25.8 / 0.65 / 0.20 |
| | 2nd Ralston | 26.6 / 0.68 / 0.18 | 26.1 / 0.67 / 0.18 | 25.9 / 0.67 / 0.20 | 26.8 / 0.71 / **0.17** | 25.8 / 0.65 / 0.20 |

† Ancestral sampling of the Markov model that discretizes forward SDE (see Ho et al. (2020)).
†† See Yue et al. (2023a).
††† See Zhou et al. (2023b).

Table 10: Different samplers used in image super-resolution for diffusion bridges. We measure their effectiveness using PSNR / SSIM / LPIPS metrics for Network Forward Evaluations (NFE) = 5, 35, and 100. Best results across samplers are **bolded** and second best are underscored. Ancestral sampling remains the best choice globally. Note that no deterministic sampler achieves an LPIPS below 0.3, even with 100 NFEs.

| NFE | Sampler | BBDM | DDBM-VE | DDBE-VP | I$^2$SB | GOUB | UniDB |
|---|---|---|---|---|---|---|---|
| 5 | Euler ODE | 28.5 / **0.77** / 0.36 | 28.6 / **0.78** / 0.31 | 28.7 / **0.79** / 0.30 | 28.8 / 0.78 / 0.34 | 28.7 / 0.77 / 0.34 | 29.2 / **0.79** / 0.32 |
|  | Euler SDE | 27.4 / 0.72 / **0.25** | 14.0 / 0.05 / 1.22 | 11.6 / 0.03 / 1.35 | 23.2 / 0.31 / 0.74 | 17.6 / 0.10 / 0.89 | 13.2 / 0.09 / 0.86 |
|  | Ancestral† | 28.1 / 0.75 / 0.28 | 28.5 / 0.77 / **0.24** | 28.8 / 0.78 / **0.23** | 28.7 / 0.77 / **0.25** | 28.5 / 0.77 / **0.27** | 28.9 / 0.78 / **0.25** |
|  | EI-ODE | 3.4 / -0.37 / 0.86 | 2.6 / -0.43 / 0.83 | 2.6 / -0.43 / 0.83 | 2.9 / -0.41 / 0.84 | 12.8 / 0.48 / 0.60 | 12.9 / 0.49 / 0.59 |
|  | Mean-ODE†† | 28.5 / **0.77** / 0.34 | 25.8 / 0.74 / 0.34 | 27.1 / 0.77 / 0.32 | 29.0 / **0.79** / 0.31 | 28.3 / 0.77 / 0.32 | 28.9 / 0.78 / 0.31 |
|  | Langevin-Heun††† | **28.7** / 0.77 / 0.35 | **29.2** / **0.79** / 0.30 | **29.4** / **0.79** / 0.28 | **29.2** / **0.79** / 0.30 | **29.0** / 0.78 / 0.31 | **29.4** / **0.79** / 0.29 |
|  | 2nd Heun | 11.6 / 0.22 / 0.73 | 16.9 / 0.19 / 0.72 | 12.9 / 0.41 / 0.59 | 9.9 / 0.08 / 0.73 | 9.9 / 0.15 / 0.74 | 11.4 / 0.24 / 0.69 |
|  | 2nd Midpoint | 28.6 / **0.77** / 0.35 | 27.1 / 0.75 / 0.33 | 21.8 / 0.74 / 0.33 | 28.8 / 0.78 / 0.35 | 27.0 / 0.73 / 0.38 | 27.4 / 0.75 / 0.34 |
|  | 2nd Ralston | 28.4 / 0.76 / 0.36 | 27.9 / 0.75 / 0.33 | 21.5 / 0.72 / 0.36 | 28.4 / 0.77 / 0.38 | 26.9 / 0.72 / 0.45 | 27.3 / 0.73 / 0.42 |
| 35 | Euler ODE | **28.5** / 0.77 / 0.35 | 27.8 / **0.77** / 0.32 | **28.5** / 0.78 / 0.31 | **28.9** / 0.78 / 0.31 | **28.6** / 0.77 / 0.32 | **29.3** / **0.79** / 0.31 |
|  | Euler SDE | 27.2 / 0.73 / 0.26 | 24.2 / 0.39 / 0.43 | 23.0 / 0.31 / 0.53 | 27.1 / 0.70 / **0.18** | 26.4 / 0.67 / **0.20** | 25.5 / 0.56 / 0.31 |
|  | Ancestral | 27.1 / 0.73 / **0.25** | 27.1 / 0.72 / **0.19** | 27.5 / 0.74 / **0.19** | 27.5 / 0.73 / **0.18** | 27.2 / 0.73 / 0.22 | 27.6 / 0.74 / **0.18** |
|  | EI-ODE | 10.6 / 0.24 / 0.71 | 12.5 / 0.35 / 0.70 | 13.1 / 0.36 / 0.68 | 4.3 / -0.25 / 0.81 | 15.5 / 0.42 / 0.68 | 11.9 / 0.26 / 0.67 |
|  | Mean-ODE | 28.2 / 0.76 / 0.34 | 24.3 / 0.73 / 0.36 | 26.3 / 0.76 / 0.34 | 25.5 / 0.75 / 0.34 | 27.8 / 0.76 / 0.33 | 28.9 / 0.78 / 0.32 |
|  | Langevin-Heun | 14.7 / 0.08 / 0.92 | 21.7 / 0.27 / 0.46 | 19.5 / 0.19 / 0.56 | 19.6 / 0.17 / 0.93 | 23.4 / 0.45 / 0.31 | 23.6 / 0.46 / 0.36 |
|  | 2nd Heun | 25.7 / 0.70 / 0.39 | 27.6 / 0.76 / 0.32 | 27.9 / 0.76 / 0.31 | 27.3 / 0.75 / 0.32 | 25.9 / 0.71 / 0.36 | 27.5 / 0.75 / 0.32 |
|  | 2nd Midpoint | **28.5** / 0.76 / 0.34 | 27.5 / **0.77** / 0.32 | 28.4 / **0.78** / 0.31 | 28.5 / **0.78** / 0.31 | 28.5 / **0.77** / 0.32 | 29.2 / **0.79** / 0.31 |
|  | 2nd Ralston | 28.4 / 0.76 / 0.35 | 27.6 / **0.77** / 0.32 | 28.4 / **0.78** / 0.31 | 28.6 / **0.78** / 0.31 | 28.5 / **0.77** / 0.32 | 29.2 / **0.79** / 0.31 |
| 100 | Euler ODE | **28.4** / 0.76 / 0.34 | 27.5 / **0.77** / 0.32 | **28.4** / 0.78 / 0.31 | **28.2** / 0.78 / 0.31 | **28.5** / 0.77 / 0.32 | **29.2** / **0.79** / 0.31 |
|  | Euler SDE | 26.9 / 0.72 / 0.25 | 25.6 / 0.56 / 0.28 | 25.3 / 0.51 / 0.32 | 26.9 / 0.71 / **0.17** | 26.4 / 0.70 / **0.21** | 26.3 / 0.66 / 0.20 |
|  | Ancestral | 26.8 / 0.72 / **0.24** | 26.6 / 0.70 / **0.18** | 27.0 / 0.72 / **0.18** | 27.0 / 0.72 / **0.17** | 26.4 / 0.71 / **0.21** | 26.3 / 0.70 / **0.18** |
|  | EI-ODE | 21.7 / 0.68 / 0.48 | 14.7 / 0.54 / 0.53 | 16.5 / 0.60 / 0.48 | 12.1 / 0.47 / 0.63 | 16.0 / 0.45 / 0.66 | 13.8 / 0.33 / 0.64 |
|  | Mean-ODE | 28.1 / **0.76** / 0.34 | 24.1 / 0.73 / 0.37 | 26.1 / 0.75 / 0.34 | 24.1 / 0.74 / 0.36 | 27.6 / 0.76 / 0.34 | 28.9 / 0.78 / 0.32 |
|  | Langevin-Heun | 27.2 / 0.69 / 0.37 | 25.8 / 0.62 / 0.24 | 25.7 / 0.58 / 0.27 | 26.8 / 0.67 / 0.26 | 25.9 / 0.69 / **0.21** | 26.0 / 0.66 / 0.20 |
|  | 2nd Heun | 27.7 / 0.75 / 0.36 | 27.4 / **0.77** / 0.33 | 28.3 / 0.78 / 0.31 | 27.8 / 0.77 / 0.32 | 27.9 / 0.76 / 0.33 | 28.9 / 0.78 / 0.31 |
|  | 2nd Midpoint | **28.4** / 0.76 / 0.34 | 27.3 / **0.77** / 0.32 | 28.3 / **0.78** / 0.31 | 27.7 / **0.78** / 0.32 | **28.5** / 0.77 / 0.32 | 29.2 / **0.79** / 0.31 |
|  | 2nd Ralston | **28.4** / 0.76 / 0.34 | 27.3 / **0.77** / 0.32 | 28.3 / **0.78** / 0.31 | 27.9 / **0.78** / 0.32 | 28.4 / **0.77** / 0.32 | 29.2 / **0.79** / 0.31 |

† Ancestral sampling of the Markov model that discretizes forward SDE (see Ho et al. (2020)).
†† See Yue et al. (2023a).
††† See Zhou et al. (2023b).

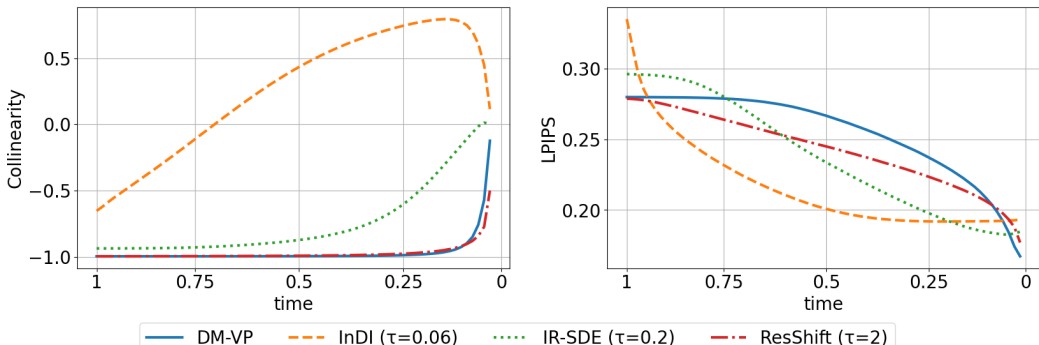

Figure 5: Comparison of diffusion and three OU processes: InDI, IR-SDE, and ResShift with progressively higher temperature parameter $\tau$, to study how temperature influences the collinearity of $\mathbf{x}_t, \mathbf{y}, \hat{\mathbf{x}}_{0|t}$, and how this collinearity, in turn, affects the LPIPS metric. Left: collinearity of normalized vectors $\mathbf{x}_t - \mathbf{y}$ and $\hat{\mathbf{x}}_{0|t} - \mathbf{x}_t$. InDI, with the lowest $\tau$, is trained to extrapolate the $\mathbf{x}_t - \mathbf{y}$ vector, especially at the end of the trajectory, where the transition variance was very small during training. Right: LPIPS of predicted $\hat{\mathbf{x}}_{0|t}$ for different $t$. It can be noticed that when InDI has high collinearity, it does not improve over time. Later steps are redundant or even harmful to the quality of the final sample. Experiments were conducted on image super-resolution.

Table 11: Different discretization strategies (see Figure 4) for image super-resolution. The results show that model quality does not depend on the chosen method. We report PSNR / SSIM / LPIPS. The best values are **bolded** and second to best are underscored.

| NFE | Method | Linear | Karras† | DDBM†† |
|---|---|---|---|---|
| 5 | DM-VP | 29.0 / 0.78 / 0.23 | 28.9 / 0.77 / **0.20** | **29.4 / 0.79** / 0.26 |
| | FM | 28.7 / 0.78 / **0.24** | 28.9 / 0.78 / 0.24 | **29.1 / 0.79** / 0.27 |
| | ResShift | 28.8 / **0.78** / 0.22 | 28.8 / 0.77 / **0.21** | **29.0 / 0.78** / 0.23 |
| | IR-SDE | 28.6 / 0.77 / **0.25** | 28.9 / 0.78 / 0.27 | 28.9 / 0.78 / 0.28 |
| | InDI | 28.6 / 0.77 / 0.32 | **28.7 / 0.77** / 0.33 | 28.5 / 0.77 / **0.31** |
| | BBDM | 28.1 / 0.75 / **0.28** | 28.1 / 0.75 / 0.28 | **28.4 / 0.76** / 0.30 |
| | DDBM-VE | 28.5 / 0.77 / 0.24 | **28.8 / 0.77** / 0.25 | 28.8 / 0.78 / 0.25 |
| | DDBM-VP | 28.8 / 0.78 / 0.23 | 29.0 / 0.78 / 0.24 | **29.1 / 0.79** / 0.25 |
| | I²SB | 28.7 / 0.77 / 0.25 | **28.9 / 0.78** / 0.26 | 28.8 / 0.78 / 0.25 |
| | GOUB | 28.5 / 0.77 / 0.27 | 28.7 / 0.77 / 0.28 | **28.9 / 0.78** / 0.30 |
| | UniDB | 28.9 / 0.78 / 0.25 | 29.1 / 0.78 / 0.26 | **29.3 / 0.79** / 0.27 |
| 35 | DM-VP | **27.9 / 0.75** / 0.18 | 27.3 / 0.71 / **0.17** | 27.8 / 0.73 / **0.17** |
| | FM | 27.2 / 0.73 / 0.18 | 27.1 / 0.71 / 0.17 | **27.7 / 0.74 / 0.17** |
| | IR-SDE | 27.0 / 0.72 / 0.18 | 27.2 / 0.72 / 0.18 | **27.9 / 0.75** / 0.20 |
| | ResShift | 27.7 / **0.75** / 0.18 | 27.5 / 0.73 / 0.17 | **27.8 / 0.74 / 0.17** |
| | InDI | 27.2 / 0.71 / **0.20** | **27.8 / 0.74** / 0.25 | 27.1 / 0.73 / 0.23 |
| | BBDM | **27.1 / 0.73** / 0.25 | 26.9 / 0.72 / 0.24 | **27.1** / 0.72 / 0.25 |
| | DDBM-VE | 27.1 / 0.72 / 0.19 | 27.3 / 0.72 / 0.17 | **27.5 / 0.73 / 0.17** |
| | DDBM-VP | 27.5 / **0.74** / 0.19 | 27.4 / 0.72 / **0.16** | **27.8 / 0.74** / 0.17 |
| | I²SB | 27.5 / 0.73 / **0.18** | **27.7 / 0.73 / 0.18** | 27.6 / 0.74 / 0.18 |
| | GOUB | 27.2 / 0.73 / **0.22** | 27.4 / 0.74 / 0.23 | **27.8 / 0.75** / 0.24 |
| | UniDB | 27.6 / 0.74 / 0.18 | 27.9 / 0.74 / 0.18 | **28.4 / 0.76** / 0.20 |
| 100 | DM-VP | **27.2 / 0.72 / 0.16** | 26.7 / 0.68 / 0.19 | 27.0 / 0.69 / 0.18 |
| | FM | 26.5 / 0.70 / 0.17 | 26.2 / 0.68 / 0.18 | **26.7 / 0.70 / 0.17** |
| | IR-SDE | 26.2 / 0.69 / 0.19 | 26.2 / 0.69 / 0.19 | **26.7 / 0.71 / 0.18** |
| | ResShift | **27.2 / 0.73** / 0.17 | 27.1 / 0.72 / 0.17 | **27.2** / 0.72 / **0.17** |
| | InDI | 26.4 / 0.68 / **0.19** | **26.9 / 0.71** / 0.21 | 26.1 / 0.69 / 0.20 |
| | BBDM | **26.8 / 0.72** / 0.24 | 26.6 / 0.71 / 0.24 | **26.7 / 0.71** / 0.24 |
| | DDBM-VE | 26.6 / 0.70 / 0.18 | 26.7 / 0.69 / 0.18 | **26.8 / 0.70 / 0.17** |
| | DDBM-VP | **27.0 / 0.72** / 0.18 | 26.8 / 0.70 / 0.17 | **27.0 / 0.71 / 0.16** |
| | I²SB | 27.0 / 0.72 / 0.17 | **27.2 / 0.71 / 0.17** | 27.0 / 0.71 / 0.17 |
| | GOUB | 26.4 / 0.71 / **0.21** | 26.7 / 0.72 / 0.22 | 16.0 / 0.41 / 0.53 |
| | UniDB | 26.3 / 0.70 / 0.18 | **27.2 / 0.71 / 0.17** | 13.3 / 0.23 / 0.67 |

† See Karras et al. (2022) Appendix D.1.
†† See Zhou et al. (2023b). Formulation is taken from the official Github repository.

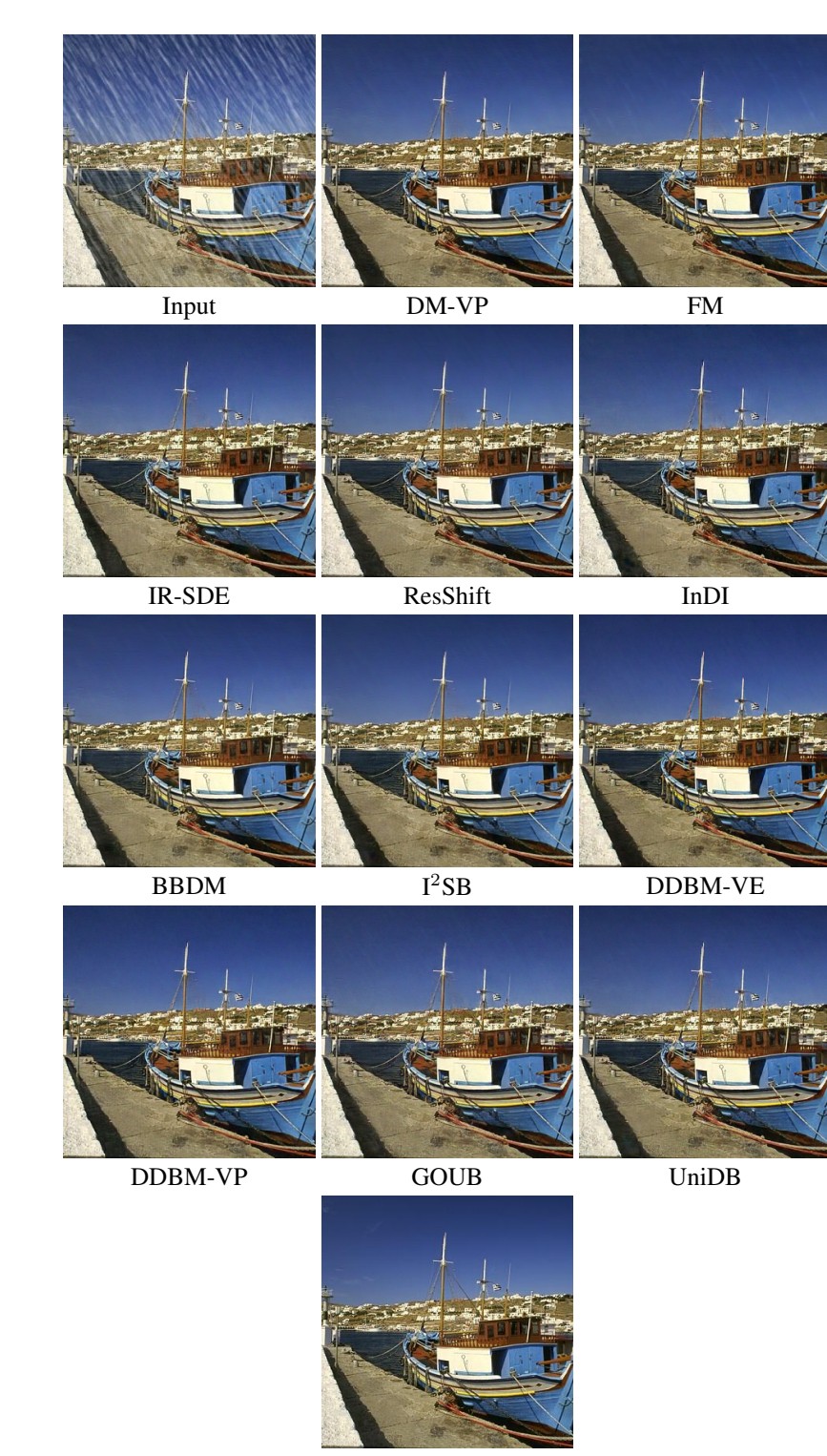

Figure 6: Visual comparison of image deraining on the Rain1400 dataset using ancestral sampling with 35 steps. All outputs appear similar, but diffusion and flow matching tend to leave slightly more visible traces of raindrops.

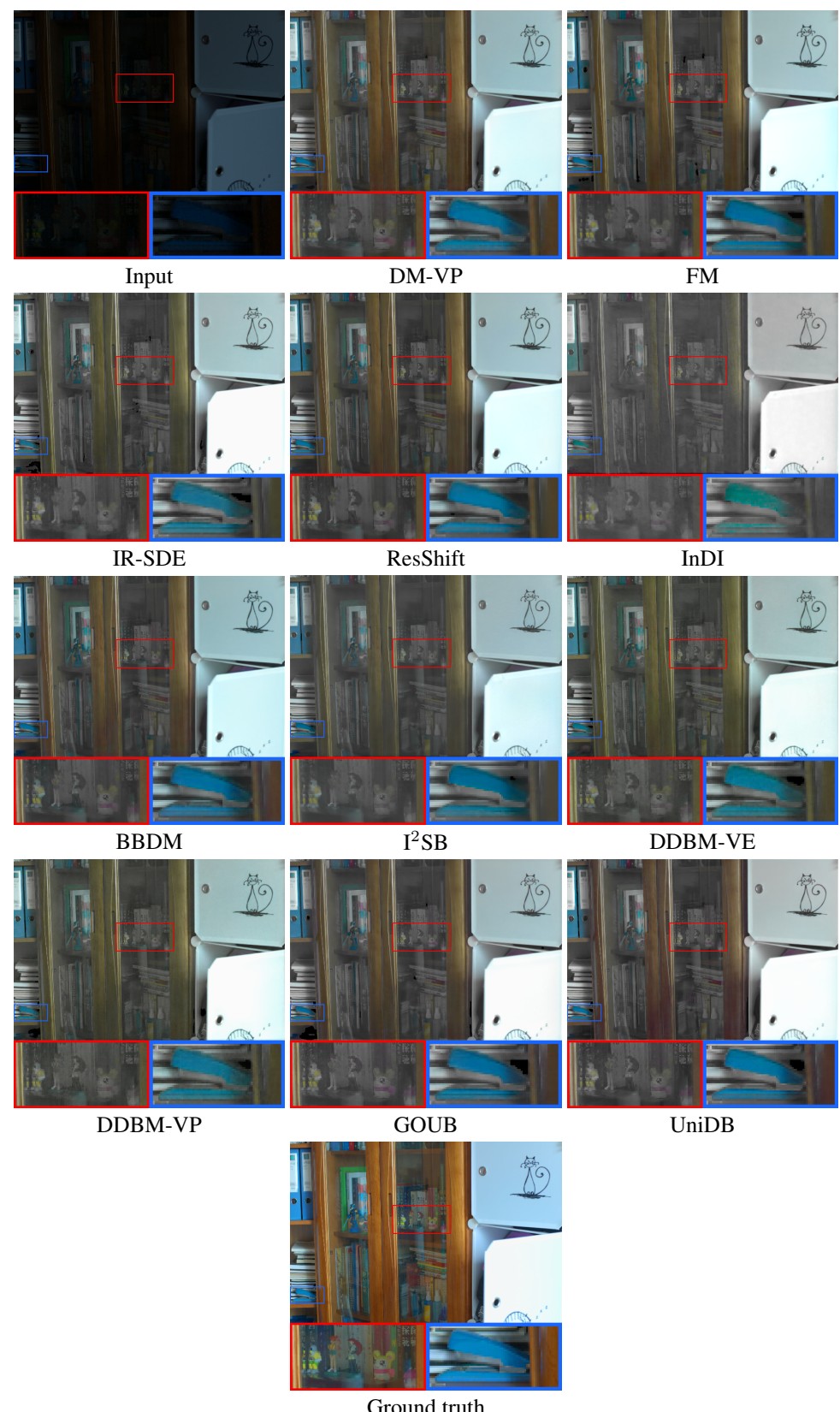

Figure 7: Visual comparison of low-light image enhancement on the LOL dataset using ancestral sampling with 35 steps. Most methods produce images of similar quality, while InDI performs noticeably worse. All methods still struggle to fully recover the ground truth.

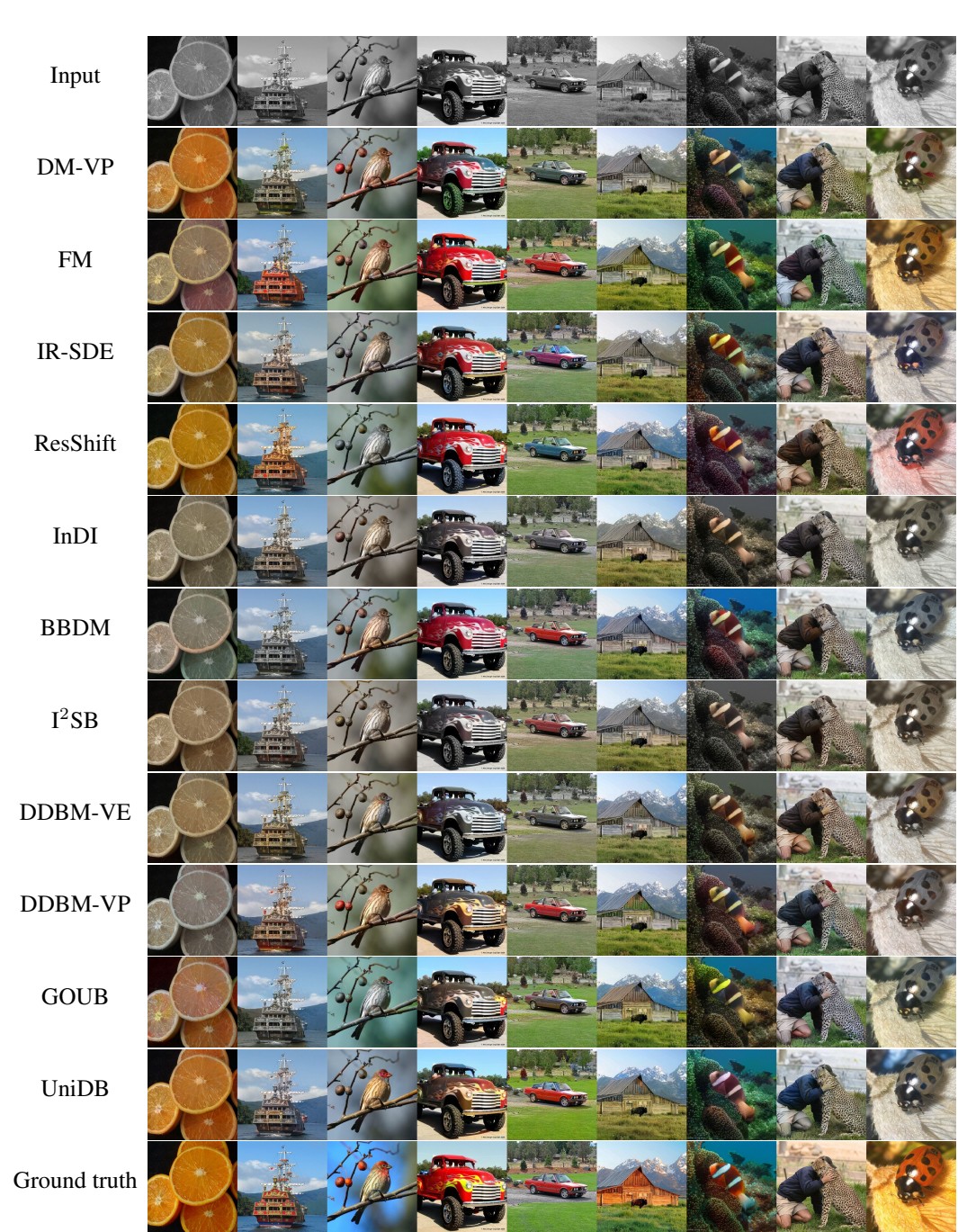

Figure 8: Visual comparison of image colorization on ImageNet. Results were generated using ancestral sampling with 35 steps. In most cases, methods recover the correct canonical colors (e.g., green grass, blue sky, orange fruit). However there are some exceptions, such as DDPM producing green tires and most methods struggling with colors of the ladybug. Flow Matching and ResShift produces outputs with higher saturation.

