# OpenReview forum: "Elucidating the design space of deep stochastic processes for image enhancement"
_ICLR.cc/2026/Conference — Submitted to ICLR 2026_

### Official Review · Reviewer_iVn3 · 2025-10-16

**Soundness:** 3
**Presentation:** 3
**Contribution:** 1
**Rating:** 2
**Confidence:** 4

**Summary:**

This paper focuses on deep stochastic processes for image enhancement, including Ornstein-Uhlenbeck processes, diffusion bridges, and diffusion processes. This paper unifies them into a single framework and develops a PyTorch library for easy implementation. The authors claim that most methods perform similarly and plain diffusion often outperforms other complex variants.

**Strengths:**

1. The paper coherently categorizes 11 existing deep stochastic process methods for image enhancement into three classes and clarifies core design differences between approaches.
2. The authors developed a PyTorch library that supports easy implementation, lowering barriers for future research.
3. This paper identifies specific performance-limiting factors (e.g., low temperature harming OU process performance).

**Weaknesses:**

1. The paper reads more like a verification report of existing methods rather than an original research contribution. While it reproduces and compares several known techniques, it fails to clarify which specific theoretical differences among these methods lead to their varying performance. A deeper analysis of the underlying theoretical factors that account for the observed performance gaps would make the contribution more insightful and meaningful.
2. The work lacks guiding insights or implications for future research directions. There is little discussion on how the presented results could inspire further development or improvement of current methodologies.
3. The paper does not introduce any novel theoretical framework or superior implementation strategy that advances the SOTA.
4. The experimental findings, for example, the observation that a plain diffusion model outperforms more complex variants, are interesting, but they are not supported by sufficient theoretical analysis or explanation.

**Questions:**

1. Could the authors clarify the key theoretical differences among the compared methods that lead to their performance variations?
2. How might the current findings provide insights or guidance for future research directions?
3. Can the authors offer a theoretical explanation for why the plain diffusion model outperforms more complex variants? Does the amount of training data or training scale have an impact?

---

> ### Author Response · Authors · 2025-11-18
>
> Thank you for your review.
>
> **Weakness 1: The paper reads more like a verification report.**
>
> We respectfully disagree that this work is only a review. Our main contribution is the unification and generalization of existing interpolation-based methods within a single continuous framework.
>
> (i) For discrete Markov chain–based methods, **Proposition 2.2** introduces a continuous stochastic process that matches their marginal distributions at the specified time steps, while also allowing evaluation at any intermediate time for greater flexibility.
>
> (ii) For flow-based models, through **Propositions 2.1** and **2.3** we are the first to define an SDE whose reverse ODE matches the original flow model in distribution and, under specific solvers, also in trajectory.
>
> (iii) Additionally, in **Remarks 2.1** and **2.2** we show that models derived from Brownian and Schrödinger Bridges can also be defined using Doob’s h-transform.
>
> **Weakness 3: Paper does not provide new state-of-the-art method.**
>
> We hold a different perspective on research, viewing it as a process of discovering method's fundamental properties and their impact on the results rather than necessarily introducing a new state-of-the-art. Our findings provide several benefits to the field:
>
> (i) it clarifies the true differences between method definitions,
>
> (ii) it allows fair comparison using the same solvers and discretization techniques, and explains why some methods perform better than others (e.g., issues with low temperature discussed in the paper),
>
> (iii) it provides unified source code, which can support further improvements in the field,
>
> (iv) We note that, without such unification, prior works rarely compare methods directly. For example, ResShift [1] acknowledges methods like InDI [2] but treats them as separate due to different formulations, and in GOUB [3], the methods BBDM [4] and I2SB [5], although cited as instances of Brownian and Schrödinger Bridges, were never directly compared to other diffusion bridge methods, as they were considered part of a different framework. Our contribution shows that these models can be compared fairly and, in some cases, are even equivalent (e.g. BBDM [4], DDBM-VE [6], I2SB[5]).
>
> **Weakness 1, Question 1: The paper does not provide how theoretical differences influence performance.**
>
> We thank the reviewer for the question about how the theoretical differences influence performance. We respectfully disagree that the paper does not address this point - we provide insights in the "Discussion" paragraph at line 414. We found that low variance of the marginals reduces the quality of the outputs. The low variance is directly related to low temperature of the process. We identified two reasons why low temperature leads to poorer results:
>
> (i) During training, when the temperature is low the intermediate sample $\mathbf{x}_t$ lies between the target image $\mathbf{x}_0$ and the low-quality input $\mathbf{y}$ with little stochasticity. Since the network receives $(\mathbf{x}_t, \mathbf{y}, t)$ as input, it can learn to extrapolate the vector $\mathbf{x}_t - \mathbf{y}$ with a magnitude proportional to $t$. In this scenario, the network does not properly learn the score of the target distribution but instead learns to extend previous predictions. In inference, the position of $\mathbf{x}_t$ depends on previous network outputs, so this extrapolation effect accumulates. As shown in Figure 5, low temperature increases this trend, leading to limited improvement over time (when colinearity is high, LPIPS does not decrease).
>
> (ii) With low temperature, the network effectively models only a narrow region between the two distributions. In inference, small deviations from this region (caused by model imperfections) push the trajectory into areas that it has not learned well. Because there is little noise to mask these imperfections, errors accumulate as the sampling progresses.
>
> **Weakness 4, Question 3: Why diffusion can outperform more complex variants?**
>
> Based on the findings from the previous point, it is clear that diffusion (and flow matching) do not have this problem because the intermediate sample $\mathbf{x}_t$ does not depend on the $\mathbf{y}$, as these processes are unconditional. Because $\mathbf{x}_t$ and $\mathbf{y}$ are independent, network cannot learn to extrapolate and is not biased toward its previous predictions during the inference. Additionally, in this case the forward process diffuses the initial condition $\mathbf{x}_0$ almost uniformly in all directions (rather than toward the corresponding $\mathbf{y}$). This neglects the second issue with the risk of out-of-distribution trajectories, since the entire space is modeled uniformly.

---

> > ### Author Response · Authors · 2025-11-18
> >
> > **Weakness 2, Question 2: What are the possible future directions from this work?**
> >
> > The experimental results suggest that the expected advantage of conditional processes such as Ornstein–Uhlenbeck and bridge models (namely, focusing on a smaller space to capture more detail) can in practice become a limitation, and our findings offer a new perspective on the inherent flaws of these models. Future work will need to take this into account. One possible direction is to train with a higher temperature while sampling with a lower one, although it remains unclear whether this approach can match the realism and diversity achieved by diffusion and flow-matching methods under the same training conditions. Another possible direction is to remove $\mathbf{y}$ from the network input, although this would limit the model’s ability to remain faithful to the conditioning.
> >
> > We will add these conclusions at the paper.
> >
> > **References:**
> >
> > [1] Zongsheng Yue, Jianyi Wang, and Chen Change Loy. Resshift: Efficient diffusion model for image
> > super-resolution by residual shifting. Advances in Neural Information Processing Systems, 36:
> > 13294–13307, 2023b.
> >
> > [2] Mauricio Delbracio and Peyman Milanfar. Inversion by direct iteration: An alternative to denoising
> > diffusion for image restoration. arXiv preprint arXiv:2303.11435, 2023.
> >
> > [3] Conghan Yue, Zhengwei Peng, Junlong Ma, Shiyan Du, Pengxu Wei, and Dongyu Zhang. Image restoration through generalized ornstein-uhlenbeck bridge. arXiv preprint arXiv:2312.10299,
> > 2023a.
> >
> > [4] Bo Li, Kaitao Xue, Bin Liu, and Yu-Kun Lai. Bbdm: Image-to-image translation with brownian
> > bridge diffusion models. In Proceedings of the IEEE/CVF conference on computer vision and
> > pattern Recognition, pp. 1952–1961, 2023.
> >
> > [5] Guan-Horng Liu, Arash Vahdat, De-An Huang, Evangelos A Theodorou, Weili Nie, and Anima
> > Anandkumar. Iˆ2 sb: Image-to-image schrodinger bridge. arXiv preprint arXiv:2302.05872,
> > 2023.
> >
> > [6] Linqi Zhou, Aaron Lou, Samar Khanna, and Stefano Ermon. Denoising diffusion bridge models.
> > arXiv preprint arXiv:2309.16948, 2023b.

---

> > > ### Comment · Reviewer_iVn3 · 2025-11-24
> > > **Response**
> > >
> > > Thank you very much for the rebuttal. It addresses part of my earlier questions, but several concerns remain.
> > >
> > > First, while I acknowledge the observation that low marginal variance degrades output quality, as noted in the strengths section, the paper does not provide empirical evidence showing how output quality or evaluation metrics vary as a function of temperature. Presenting explicit ablations, such as curves or tables that directly illustrate metric changes under different temperature settings, would substantially strengthen this claim.
> > >
> > > Second, the paper does not analyze how the specific modeling choices of different methods impact performance. For instance, IR-SDE, ResShift, and InDI differ considerably in their concrete implementations. If these methods are configured to use the same settings, how would their performance differ? This lack of analysis limits our ability to understand how particular modeling decisions affect final performance.
> > >
> > > Overall, I believe that the sufficiency and completeness of the experimental study still require improvement.

---

### Official Review · Reviewer_rTk4 · 2025-10-20

**Soundness:** 3
**Presentation:** 3
**Contribution:** 3
**Rating:** 6
**Confidence:** 5

**Summary:**

The paper unifies recent “stochastic process” approaches in image enhancement (diffusion, OU, bridge) under a single SDE framework, provides a unified table and three representative propositions, and includes FM, ResShift, and InDI as special cases within specific SDE formulations. It reports systematic comparisons across four tasks—super-resolution, upsampling, low-light enhancement, colorization, and deraining—concluding that conditional stochastic processes do not significantly outperform standard diffusion. The authors also release a modular codebase to facilitate implementation with different process paths. The goals and motivations are clear, and the work offers strong value in framework unification.

**Strengths:**

1. **Clear unified perspective and standardized presentation:** The paper systematically formulates these image enhancement methods in a general form, decoupled from the scheduler/sampler, facilitating fair comparison and reproducibility.
2. **Comprehensive empirical baselines:** It covers DM-VP, FM, IR-SDE, ResShift, InDI, (BBDM/DDBM/I2SB/GOUB/UniDB), unifies the number of steps, and reports PSNR/SSIM/LPIPS across multiple tasks. The finding that most methods perform similarly provides valuable insight for the community.
3. **Engineering contribution:** A unified, extensible PyTorch library is provided, supporting replication and modular experimentation.

**Weaknesses:**

**Experimental Setting:**

1. To ensure fairness, the authors unified network architectures, loss functions, parameterizations, and sampling formulas across methods—leading to the conclusion that performance differences are minor. However, this may actually introduce a different kind of unfairness. Each method may require distinct hyperparameters or architectures to reach its theoretical potential. Using one universal configuration or directly adopting settings from prior work could prevent methods from achieving their best possible performance. For example, in Table 2 (deraining) and Tables 7–8 (NFE=100), UniDB performs notably worse. Comparing methods’ theoretical merits should involve finding their optimal performance points rather than enforcing universal parameters, as even DDPM might fail to outperform an autoencoder under such constraints.
2. The paper lacks details on the parameterization strategies used in training. These methods typically fall into three categories: (1) parameterizing the inverse probability $p_\theta(x_0|x_t)$ (essentially $\epsilon_\theta$), (2) parameterizing the velocity field $v_\theta(x_t)$, and (3) SDE-based parameterization. The parameterization type strongly affects the sampling process. For instance, if SDE-based parameterization is used, Mean-ODE tends to perform well, while Euler-ODE does not.

**Questions:**

1. Regarding Weakness 1, it is recommended that the authors tune hyperparameters more thoroughly to find each method’s optimal configuration and update the main experimental tables accordingly. Although this might not be feasible during the rebuttal stage, such revisions should be considered for the camera-ready version or future submissions, along with a discussion of this issue in the experimental setup.
2. Regarding Weakness 2, could the inconsistent parameterization schemes have caused the anomalies observed in Table 9, where several results are abnormally low? If not, the authors should provide an explanation.

Finally, I believe that differences among these methods do exist. Even from Table 2, one can see that bridge-based models perform mediocrely in super-resolution—worse than DM-VP—but significantly outperform both diffusion and OU models in colorization. With more datasets and fairer comparisons, more universal findings are likely to emerge.

---

> ### Author Response · Authors · 2025-11-18
> **Response**
>
> Thank you for your review.
>
> **Weakness 1, Question 1: Standardized values of hyperparameters might introduce unfairness.**
>
> We agree that other factors may affect the performance of the methods. In this work, we followed the rationale that, in order to study the influence of process definition on quality, all other components should remain fixed. We adopted this approach because process definition is typically the main contribution of the considered methods. However, as noted, each method also includes additional design choices such as the parameterization of the network or the loss function, which can affect the results. Exploring every possible configuration would require an extensive grid search. Therefore, we will follow your advice and begin by comparing implementations with their original hyperparameter values to verify whether we can observe any synergies that may offer an advantage over our standardized choices.
> However, we would like to exclude the use of the original backbone architecture, as this would violate our intention to maintain a fair comparison of processes independent of the backbone. For this reason, we focus on aspects such as network parameterization and the loss function shown in Table 3.
>
> Training for image super-resolution on FFHQ with the original parameters is already in progress, and we will provide an update within a few days. We will make every effort to report additional experimental results from other tasks and datasets before the end of the rebuttal stage so that we can discuss the results; otherwise, they will be included in the revised version.
>
> **Weakness 2: Used parametrization strategy.**
>
> We used the same network parameterization ($\hat{x}_0$ prediction) for all methods, as our preliminary experiments showed it performs best overall. We stated this fact in **Experimental setup** subsection. However, as mentioned previously, in the revised version we will include experiments using the original parameterization, allowing us to assess its impact on different solvers during inference.
>
> **Question 2: Lower metrics in Table 9 for some methods.**
>
> Without empirical results, we cannot be certain that other combinations of method, parameterization, and solver would behave differently. However, it is likely that the lower scores in Table 9 can be explained in another way.
>     We divide the explanation into two cases: (i) FM [1] and InDI [2] methods, and (ii) the second-order Heun solver with low NFE (Number of Feedforward Evaluations).
>
> (i) Flow Matching and InDI [2] have stiff dynamics. Unlike Diffusion [3,4], IR-SDE [5], or ResShift [6], which only approximate the base distribution at $t=1$, flow-based methods are designed to converge exactly to $p_{base}$ at $t=1$. As a result, our propositions have drift and diffusion coefficients that tend toward $-\infty$ and $\infty$, respectively, when we approach $t=1$.
> This can cause difficulties for some solvers, while others remain stable.
> For example, for the Euler-ODE solver, these terms cancel out, which reproduces the sampling procedure from the original work. 2nd order Runge–Kutta solvers behave similarly, since they are built on Euler steps. Ancestral sampling also remains stable because it is based on the transition kernel $p(\mathbf{x}_t | \mathbf{x}_s)$. However, other solvers become unstable, leading to the observed failures.
>
> (ii) To maintain the same number of function evaluations (NFE), higher-order solvers, such as second-order Runge-Kutta, perform fewer steps. For example, with NFE = 5, the Heun method performs two Heun steps and one final evaluation. This setup is consistent with prior works. However, with such a low NFE, the state is updated only three times. Because Heun averages (equally) two Euler steps, it tends to sample from higher $t$ values, while Ralston and Midpoint give more weight to the second Euler step when $t$ is lower. This makes Heun more sensitive to low NFE.

---

> ### Author Response · Authors · 2025-11-18
>
> **References:**
> (Separately - due to the character limits)
>
> [1] Xingchao Liu, Chengyue Gong, and Qiang Liu. Flow straight and fast: Learning to generate and
> transfer data with rectified flow. arXiv preprint arXiv:2209.03003, 2022.
>
> [2] Mauricio Delbracio and Peyman Milanfar. Inversion by direct iteration: An alternative to denoising
> diffusion for image restoration. arXiv preprint arXiv:2303.11435, 2023.
>
> [3] Jonathan Ho, Ajay Jain, and Pieter Abbeel. Denoising diffusion probabilistic models. Advances in
> neural information processing systems, 33:6840–6851, 2020.
>
> [4] Yang Song, Jascha Sohl-Dickstein, Diederik P Kingma, Abhishek Kumar, Stefano Ermon, and Ben
> Poole. Score-based generative modeling through stochastic differential equations. arXiv preprint
> arXiv:2011.13456, 2020b.
>
> [5] Ziwei Luo, Fredrik K Gustafsson, Zheng Zhao, Jens Sj¨olund, and Thomas B Schon. Image restoration with mean-reverting stochastic differential equations. arXiv preprint arXiv:2301.11699,
> 2023a.
>
> [6] Zongsheng Yue, Jianyi Wang, and Chen Change Loy. Resshift: Efficient diffusion model for image
> super-resolution by residual shifting. Advances in Neural Information Processing Systems, 36:
> 13294–13307, 2023b.

---

> > ### Comment · Reviewer_rTk4 · 2025-11-24
> > **Final Comment**
> >
> > I have read the authors' rebuttal, and it largely addresses my concerns. I hope the authors will update the final version of the paper with the fairer experimental results (or provide a clear explanation) as discussed.
> >
> > Based on this, I maintain my score.

---

### Official Review · Reviewer_aiws · 2025-10-29

**Soundness:** 2
**Presentation:** 2
**Contribution:** 2
**Rating:** 2
**Confidence:** 4

**Summary:**

This paper integrates 11 methods of SDEs  in a systematic table and proposes an unified mathematical framework for definition to present the stochastic processes. They also provide a modular library (so far information about this library is only limited to description provided in this paper) that implements the methods and evaluate them on four image enhancement tasks: super-resolution, colorization, low-light enhancement to improve comparison.

**Strengths:**

This paper serves more like a review paper to integrate SDEs diffusion methods in a systematic table. They also provide a modular library. This seems like will bring out convivence for implementing related methods.
modular library

**Weaknesses:**

This paper serves more like a review paper.
1., the introduction section ignored some important events when reviewing literatures.
In P1. Line 50, 4th paragraph, they should have referred to papers on diffusion models.
2. I would suggest to add reference links to the table to make it easier for readers to access the references.

**Questions:**

1. I wonder what is the benefit of this unified framework.
2. Did you choose specific parameters to represent in the table? It would be beneficial to explain how you get to a formula that is different from the original paper where there is a different. For example, did you set $\theta_t=0$ in UniDB?

---

> ### Author Response · Authors · 2025-11-18
> **Response**
>
> Thank you for your review.
>
> **Weakness 1: This paper serves more like a review paper.**
>
> We respectfully disagree that this work is only a review. Our main contribution is the unification and generalization of existing interpolation-based methods within a single continuous framework.
>
> (i) For discrete Markov chain–based methods, **Proposition 2.2** introduces a continuous stochastic process that matches their marginal distributions at the specified time steps, while also allowing evaluation at any intermediate time for greater flexibility.
>
> (ii) For flow-based models, through **Propositions 2.1** and **2.3** we are the first to define an SDE whose reverse ODE matches the original flow model in distribution and, under specific solvers, also in trajectory.
>
> (iii) Additionally, in **Remarks 2.1** and **2.2** we show that models derived from Brownian and Schrödinger Bridges can also be defined using Doob’s h-transform.
>
> **Question 1: Benefits of the unified framework.**
>
> The unified mathematical framework provides several benefits:
>
> (i) it clarifies the true differences between method definitions,
>
> (ii) it allows fair comparison using the same solvers and discretization techniques, and explains why some methods perform better than others (e.g., issues with low temperature discussed in the paper),
>
> (iii) it provides unified source code, which can support further improvements in the field,
>
> (iv) We note that, without such unification, prior works rarely compare methods directly. For example, ResShift [1] acknowledges methods like InDI [2] but treats them as separate due to different formulations, and in GOUB [3], the methods BBDM [4] and I2SB [5], although cited as instances of Brownian and Schrödinger Bridges, were never directly compared to other diffusion bridge methods, as they were considered part of a different framework. Our contribution shows that these models can be compared fairly and, in some cases, are even equivalent (e.g. BBDM [4], DDBM-VE [6], I2SB [5]).
>
> **Weakness 2: Additional citations for better readability.**
>
> We thank the reviewer for suggesting additional citations in the text and Table 1. We will include them in the camera-ready version. We would like to clarify that all referenced methods are already cited in the corresponding paragraphs, so no citations were omitted, and the additional references will only help to improve readability.
>
> **Question 2: Different formulas from the original articles.**
>
> For the naming consistency across all the compared methods, we follow the notation where the noise scheduler is denoted as $\beta_t$, whereas in works such as IR-SDE [7], GOUB [3], and UniDB [8] authors used $\theta_t$. We are grateful for this comment and we will add a proper explanation in the manuscript.
>
> Regarding our propositions, they are defined as continuous stochastic processes described by SDEs, rather than discrete Markov chains or flow-based formulations from the original works. Importantly, they **generalize** (rather than modify) the original methods; with appropriate schedulers and solvers, the original implementations can be fully recovered. In our experiments, we used the original scheduler choices and tested nine different solvers (see Tables 9 and 10), including both the original solvers and alternative options.
>
> **References:**
>
> [1] Zongsheng Yue, Jianyi Wang, and Chen Change Loy. Resshift: Efficient diffusion model for image
> super-resolution by residual shifting. Advances in Neural Information Processing Systems, 36:
> 13294–13307, 2023b.
>
> [2] Mauricio Delbracio and Peyman Milanfar. Inversion by direct iteration: An alternative to denoising
> diffusion for image restoration. arXiv preprint arXiv:2303.11435, 2023.
>
> [3] Conghan Yue, Zhengwei Peng, Junlong Ma, Shiyan Du, Pengxu Wei, and Dongyu Zhang. Image restoration through generalized ornstein-uhlenbeck bridge. arXiv preprint arXiv:2312.10299,
> 2023a.
>
> [4] Bo Li, Kaitao Xue, Bin Liu, and Yu-Kun Lai. Bbdm: Image-to-image translation with brownian
> bridge diffusion models. In Proceedings of the IEEE/CVF conference on computer vision and
> pattern Recognition, pp. 1952–1961, 2023.
>
> [5] Guan-Horng Liu, Arash Vahdat, De-An Huang, Evangelos A Theodorou, Weili Nie, and Anima
> Anandkumar. Iˆ2 sb: Image-to-image schrodinger bridge. arXiv preprint arXiv:2302.05872,
> 2023.
>
> [6] Linqi Zhou, Aaron Lou, Samar Khanna, and Stefano Ermon. Denoising diffusion bridge models.
> arXiv preprint arXiv:2309.16948, 2023b.
>
> [7] Ziwei Luo, Fredrik K Gustafsson, Zheng Zhao, Jens Sj¨olund, and Thomas B Schon. Image restoration with mean-reverting stochastic differential equations. arXiv preprint arXiv:2301.11699,
> 2023a.
>
> [8] Kaizhen Zhu, Mokai Pan, Yuexin Ma, Yanwei Fu, Jingyi Yu, Jingya Wang, and Ye Shi.
> Unidb: A unified diffusion bridge framework via stochastic optimal control. arXiv preprint
> arXiv:2502.05749, 2025.

---

> > ### Comment · Reviewer_aiws · 2025-11-25
> >
> > Thanks to the authors to address my concerns. Same as reviewer  rTk4, I hope to see the final version of the paper with new experimental results and provide a clear explanation based on discussion here, and make decision there after.

---

### Official Review · Reviewer_Uwv1 · 2025-10-31

**Soundness:** 3
**Presentation:** 2
**Contribution:** 3
**Rating:** 4
**Confidence:** 3

**Summary:**

This paper addresses the fragmentation in deep stochastic process-based image enhancement (IE) methods by proposing a unified mathematical framework to consolidate 11 state-of-the-art approaches. These methods are categorized into three classes of stochastic processes—standard diffusion models, Ornstein-Uhlenbeck processes, and diffusion bridges—each formalized via SDEs, transition kernels, and base distributions.

The authors separate method definitions from their original schedulers/samplers, enabling fair comparison. They implement a modular library (Ito Vision) to standardize implementations and conduct comprehensive experiments across four IE tasks: super-resolution (FFHQ), low-light enhancement (LOL), colorization (ImageNet), and deraining (Rain1400).

Key findings include: (1) Most methods achieve comparable performance, with plain diffusion models often outperforming specialized OU/bridge methods; (2) Low-temperature settings in OU processes (e.g., InDI) lead to collinearity bias, limiting detail recovery; (3) Ancestral sampling is the most consistent sampler, while deterministic samplers cause oversmoothing for diffusion bridges.

**Strengths:**

- Foundational Unification of Fragmented Methods: The paper fills a critical gap in IE research by formalizing 11 diverse methods into a single mathematical framework (Table 1).
- The experiments set a new standard for fairness in IE comparisons with identical backbones and various tasks. Ablations on samplers (Tables 9-10) and discretization strategies (Table 11) provide actionable insights.
- The paper goes beyond surface-level comparisons to explain why some methods underperform with analysis of temperature and collinearity.

**Weaknesses:**

- The framework is validated primarily on moderate-resolution inputs (e.g., 256×256 for super-resolution, 320×320 for low-light enhancement) but lacks testing on:
  - High-resolution IE: Professional scenarios (e.g., 1K/2K image restoration) require handling larger feature maps—It is unclear if the unified framework scales to these sizes (e.g., U-Net backbones may suffer from memory constraints).
  - Real-world noise artifacts: The paper assumes synthetic degradation (e.g., bicubic downsampling for super-resolution, synthetic rain for deraining) but ignores real sensor noise (Poisson shot noise, ISP artifacts) common in low-light photography.

- All experiments use U-Net backbones, while modern IE methods increasingly adopt Transformer/DiT architectures for better long-range feature modeling. It is unclear if the unified framework’s SDE/transition kernel definitions are compatible with Transformer-based score networks. The paper does not compare U-Net with Transformer backbones under the same stochastic process, missing an opportunity to validate backbone impact.

**Questions:**

- Have you tested the unified framework on ultra-high-resolution inputs (e.g., 1024×1024) or real-world noisy images? If so, how do process performance (e.g., LPIPS) and computational cost (e.g., memory) scale? If not, what modifications (e.g., hierarchical SDEs) would be needed to support these scenarios?
- Have you explored using DiT backbones with the unified framework? If so, did you need to adjust the transition kernel definitions to accommodate token-wise attention? How do Transformer-based methods compare to U-Net in terms of performance (e.g., FID) and efficiency?

---

> ### Author Response · Authors · 2025-11-18
> **Additional experiments with high-resolution data and different backbone**
>
> Thank you for the review.
>
> **Weakness 1a, Question 1: Evaluation on higher resolution datasets.**
>
> We appreciate the suggestion to evaluate higher-resolution datasets. We are already training models on the DIV2K dataset, which provides 2K-resolution images and is widely used in the community. These experiments will be included in the revised version, but we will make every effort to report the results during the rebuttal as soon as possible.
> To handle the increased memory requirements from higher resolutions, we will employ the latent diffusion [1] framework with a pretrained VAE, similarly to what we did in image colorization on ImageNet. Hierarchical SDEs can reduce computational cost, but the final denoising stage would still encounter the same memory bottleneck as the standard diffusion process. We would also like to clarify that the resolutions of 256×256 for super-resolution and 320×320 for low-light enhancement refer only to the training crop sizes. During inference, we evaluated the full-resolution images from the datasets (512×512 and 400×600).
>
> **Weakness 1b: Evaluation on datasets with real world degradations.**
>
> Synthetic datasets alone may not fully capture real-world performance. To address this, we trained and evaluated all considered models on the Low-Light Image Enhancement (LLIE) task using the LOL dataset [2], enabling comparison under both synthetic and real-world degradations. For the synthetic datasets, we follow the standard protocol in which low-quality images are generated using established degradation procedures.
>
> **Weakness 2, Question 2: Evaluation with Transformer as the backbone.**
>
> We appreciate this valuable suggestion. The main goal of our work is to provide a unified mathematical framework for existing interpolation-based methods. At the same time, we aim to ensure fair and complete comparisons. Using a different backbone architecture introduces another important perspective, so we will include additional experiments with a transformer-based denoising function on both the image super-resolution and low-light enhancement tasks, covering synthetic and real degradations.
>
> **Question 2: Necessary adjustments to processes for different backbones.**
>
> No, this is not necessary. The stochastic processes we used are defined in the data space, not on the feature maps. Token-wise attention operates only within the network architecture and does not affect the definition of the transition kernels. We can treat the model architecture as the blackbox independent of the method definition.
>
> **References:**
>
> [1] Robin Rombach, Andreas Blattmann, Dominik Lorenz, Patrick Esser, and Bjorn Ommer. High-resolution image synthesis with latent diffusion models. In Proceedings of the IEEE/CVF conference on computer vision and pattern recognition, pp. 10684–10695, 2022.
>
> [2] Chen Wei, Wenjing Wang, Wenhan Yang, and Jiaying Liu. Deep retinex decomposition for low-light enhancement. British Machine Vision Conference, 2018.

---

### Meta-Review · Area_Chair_sXAc · 2025-12-26

**Summary:**

This submission sets out to unify the landscape of deep stochastic processes for image enhancement—consolidating 11 methods (including standard diffusion, Ornstein-Uhlenbeck processes, and diffusion bridges) into a single mathematical framework. The authors provide a standardized table of definitions, release a modular PyTorch library ("Ito Vision"), and benchmark these methods across four tasks using identical backbones to ensure a level playing field.

Ultimately, the recommendation is to reject. While the reviewers appreciate the significant engineering effort behind the "Ito Vision" library and the attempt to bring order to a fragmented field, the consensus leans towards viewing this work as a "verification report" or a survey rather than a novel research contribution. The primary finding, that plain diffusion models often outperform more complex specialized methods, is interesting, but reviewers found the theoretical explanation for why this occurs to be insufficient. Furthermore, there is a valid concern that the experimental design, which standardizes backbones and hyperparameters to ensure "fairness," may inadvertently handicap methods that rely on specific architectural choices to achieve their state-of-the-art performance.

**Reviewer Concerns:**

Addressed Concerns:

(1) Mathematical Unification: The authors clarified how their continuous SDE framework generalizes the original discrete or flow-based methods, addressing confusion regarding the theoretical grouping of these methods.

(2) Real-world Evaluation: In response to Reviewer Uwv1’s concern about reliance on synthetic degradation, the authors introduced experiments on the LOL dataset to cover real-world low-light enhancement.

(3) Tooling: The release of the codebase was universally recognized as a positive contribution to the community.

Outstanding Concerns:

(1) Contribution Type & Novelty: This was the most critical sticking point. Reviewers aiws and iVn3 maintained that the paper reads like a review or survey paper rather than original research. The unification, while valuable, was not seen by the majority as a sufficient standalone contribution without deeper theoretical novelty.

(2) Depth of Analysis: Reviewer iVn3 noted that while the authors hypothesize that "low temperature" and collinearity issues cause performance drops in OU processes, the paper lacks the empirical ablations (e.g., performance curves vs. temperature) to rigorously prove this causal link.

(3) Experimental "Fairness": Reviewer rTk4 raised a strong point that standardization might actually introduce unfairness. By forcing all methods into a universal configuration (same backbone/hyperparameters), the study may fail to capture the true potential of methods that require specific tuning, rendering the "underperformance" conclusion shaky.

(4) Scalability: Reviewer Uwv1 questioned the framework's applicability to modern, high-resolution pipelines (e.g., DiT backbones, 2K resolution). While the authors promised these for the revision, the lack of immediate results leaves the scalability of the unified framework unproven.

**Reviewer Scores:**

Reviewer Uwv1 (Score: 4->4):  The reviewer would likely have kept their score at a 4 or moved to a weak 5 at best. While they appreciated the rebuttal, their requests for high-resolution validation and Transformer backbones were met with promises for a future revision rather than immediate data, preventing a strong shift to acceptance.

Reviewer aiws (Score: 2 -> 2~4): This reviewer viewed the paper as a literature review. While the rebuttal clarified the mathematical novelty of the unification propositions, it likely wouldn't have been enough to fundamentally change the view of the paper. The reviewer explicitly stated he/she would wait for a final version with new results before making a final judgment, implying his/her skepticism remained high.

Reviewer rTk4 (Score: 6->6): This reviewer explicitly maintained their score of 6 after the rebuttal. The reviewer found value in the unification but remained wary of the experimental limitations regarding hyperparameter tuning. It is unlikely the reviewer would have advocated strongly for acceptance against the concerns of the other reviewers.

Reviewer iVn3 (Score: 2-> 2~4): This reviewer would almost certainly have kept their score at 2. In the reviewer's final comment, the reviewer explicitly stated that "several concerns remain," specifically citing the lack of empirical evidence regarding temperature/quality curves and the lack of analysis on modeling choices.

---

### Decision · Program_Chairs · 2026-01-26

Reject